# Off-Policy Evaluation of Ranking Policies for Large Action Spaces via Embeddings and User Behavior Assumption

## Abstract

Off-policy evaluation (OPE) in ranking settings with large ranking action spaces, which stems from an increase in both the number of unique actions and length of the ranking, is essential for assessing new recommender policies using only logged bandit data from previous versions. To address the high variance issues associated with existing estimators, we introduce two new assumptions: *no direct effect on rankings* and *user behavior model on ranking embedding spaces*. We then propose *the generalized marginalized inverse propensity score (GMIPS) estimator* with statistically desirable properties compared to existing ones. We characterize the parameters of our estimator that minimize the mean squared error (MSE). Finally, we demonstrate that the GMIPS achieves the lowest MSE. Notably, among GMIPS variants, the *marginalized reward interaction IPS (MRIPS)* incorporates a doubly marginalized importance weight based on a cascade behavior assumption on ranking embeddings. MRIPS effectively balances the trade-off between bias and variance, even as the ranking action spaces increase and the above assumptions may not hold, as evidenced by our experiments.

## 1. Introduction

Off-policy evaluation (OPE) in ranking settings in contextual bandits is essential for accurately assessing new recommender and retrieval policies using only logged bandit data collected from previous versions to avoid the costs associated with A/B testing (Dudík et al., 2014; Gilotte et al., 2018).

In ranking OPE, estimating the true policy value using the

conventional inverse propensity score (IPS) estimator is extremely challenging (Horvitz & Thompson, 1952). This difficulty is attributed to the exponential increase in number of rankings and the high variance problem associated with the estimator (Kiyohara et al., 2023). To address this issue, many derivatives of the IPS estimator have been developed using marginalized importance weights based on assumptions about the specific user behavior on ranking actions (Li et al., 2018; McInerney et al., 2020; Kiyohara et al., 2023). Specifically, (Li et al., 2018) developed an independent IPS (IIPS) estimator under the assumption that users reward an item in a specific position without being affected by items in other positions. (McInerney et al., 2020) developed a reward interaction IPS (RIPS) estimator under the assumption that users reward an item in a specific position after being influenced by items positioned above it. (Kiyohara et al., 2023) developed an adaptive IPS (AIPS) estimator that switches the marginalized importance weights by assuming that each user follows a behavioral distribution. The number of rankings depends on the length of the ranking and number of unique actions. To reduce the variance of the estimators with an increase in the former, one can assume a user behavior based on existing studies. However, as the latter increases, high variance may still occur, even with the previous estimators, such as IIPS, which are designed to minimize variance.

A recent study on OPE in single-action decision-making addressed the high variance issue caused by large action spaces by extending the distribution of action embeddings and clustering within the data generation process and by leveraging these techniques in the estimator (Saito & Joachims, 2022; Saito et al., 2023; Peng et al., 2023; Sachdeva et al., 2024). However, no studies have utilized action embeddings in ranking settings of OPE. In this study, we employ ranking embeddings for estimation, which extends the work of (Saito & Joachims, 2022) to ranking settings. Additionally, we introduce two new assumptions: *no direct effect on rankings*, which posits that ranking actions have no causal effect on rewards; and *user behavior model on ranking embedding spaces*, which asserts that the reward follows a specific user behavior model based on ranking embeddings. We argue that the user behavior model on ranking action spaces is also applicable to ranking embedding spaces. We then pro-

[1]Anonymous Institution, Anonymous City, Anonymous Region, Anonymous Country. Correspondence to: Anonymous Author <anon.email@domain.com>.

Preliminary work. Under review by the International Conference on Machine Learning (ICML). Do not distribute.

pose *the generalized marginalized IPS (GMIPS)* estimator, which possesses statistically desirable properties such as unbiasedness and variance reduction compared to existing estimators. We characterize the trade-off between estimator bias and variance as dependent on the degree of violation of the aforementioned two assumptions. Through experiments, we demonstrate that the GMIPS can significantly improve the mean squared error (MSE) over existing estimators under these assumptions, even as the number of unique actions and rankings increases. Experiments involving realistic scenarios where the two assumptions do not hold demonstrate that *the marginalized reward interaction IPS (MRIPS)* estimator, among GMIPS variants, which leverages the doubly marginalized importance weight and assumes a cascade behavior model on ranking embeddings, exhibits superior bias–variance control and outperforms existing estimators. Furthermore, we demonstrate that intentionally omitting certain embedding dimensions during estimation enhances the performance of MRIPS.

# 2. Problem Formulation

Here, we formulate a basic OPE framework for ranking settings. First, we introduce the necessary notation. Context $x \in \mathcal{X}$ is generated from unknown distribution $p(x)$, and policy function given context $x$ is defined as $\pi : \mathcal{X} \to \Delta(\Pi(\mathcal{A}))$, where $\mathcal{A}$ represents the unique action set and $K$ denotes the length of the ranking. Thus, we define the ranking set as $\Pi(\mathcal{A})$. Consequently, ranking action (e.g., movies) $\boldsymbol{a} := (\boldsymbol{a}(1), \cdots, \boldsymbol{a}(K)) \in \Pi(\mathcal{A})$ is generated from known distribution $\pi(\boldsymbol{a}|x) = \prod_{k=1}^{K} \pi(\boldsymbol{a}(k)|x)$. Rewards (e.g., watch time) $\boldsymbol{r} := (\boldsymbol{r}(1), \cdots, \boldsymbol{r}(K))$, given context $x$ and ranking action $\boldsymbol{a}$, are observed from unknown distribution $p(\boldsymbol{r}|x, \boldsymbol{a})$. Using these components, we define the reward function to be estimated as follows:

$$V(\pi) := \sum_{k=1}^{K} \underbrace{\mathbb{E}_{p(x)\pi(\boldsymbol{a}|x)} [q_k(x, \boldsymbol{a})]}_{V^{(k)}(\pi)} \qquad (1)$$

where $q_k(x, \boldsymbol{a}) = \mathbb{E}[\boldsymbol{r}(k)|x, \boldsymbol{a}]$ represents the expected reward for each position and $V^{(k)}(\pi)$ denotes the position-wise policy value of target policy $\pi$. We can obtain the following $n$ logged bandit data $\mathcal{D} := \{(x_i, \boldsymbol{a}_i, \boldsymbol{r}_i)\}_{i=1}^{n}$, which are distributed independently and identically.

$$(x, \boldsymbol{a}, \boldsymbol{r}) \sim p(x)\pi_0(\boldsymbol{a}|x)p(\boldsymbol{r}|x, \boldsymbol{a}) \qquad (2)$$

where $\pi_0$ is a logging policy. OPE aims to accurately estimate the policy value of target policy $\pi$ using only logged bandit data collected from logging policy $\pi_0$. It is standard practice to evaluate the MSE between true policy function $V$ and estimator $\hat{V}$.

$$\text{MSE}\left(\hat{V}(\pi; \mathcal{D})\right) = \mathbb{E}_{\mathcal{D}}\left[\left(V(\pi) - \hat{V}(\pi; \mathcal{D})\right)^2\right]$$

$$= \text{Bias}\left(\hat{V}(\pi; \mathcal{D})\right)^2 + \mathbb{V}_{\mathcal{D}}\left[\hat{V}(\pi; \mathcal{D})\right]$$

where $\text{Bias}(\hat{V}(\pi; \mathcal{D})) := V(\pi) - \mathbb{E}_{\mathcal{D}}[\hat{V}(\pi; \mathcal{D})]$ is the divergence between the true policy value and the expected value of the estimator, and $\mathbb{V}_{\mathcal{D}}[\hat{V}(\pi; \mathcal{D})] := \mathbb{E}_{\mathcal{D}}[(\hat{V}(\pi; \mathcal{D}) - \mathbb{E}_{\mathcal{D}}[\hat{V}(\pi; \mathcal{D})])^2]$ denotes the variance of the estimator. The MSE can be decomposed into the squared bias and variance terms. It is essential to consider the trade-off between bias and variance to minimize the MSE.

## 2.1. Existing Estimators

IPS is frequently employed to address the bias inherent in logging policies. In this section, we summarize the existing estimators that utilize IPS. These estimators aim to control for bias and variance by leveraging the assumptions of the user behavior model on ranking actions. We define $\hat{V} := \sum_{k=1}^{K} \hat{V}^{(k)}$ to represent the estimated value, which is the sum of position-wise effects $\hat{V}^{(k)}$. The position-wise effect of the GIPS estimator is then expressed as follows:

$$\hat{V}_{\text{GIPS}}^{(k)}(\pi; \mathcal{D}) := \frac{1}{n} \sum_{i=1}^{n} w_{\Phi_k}(x_i, \boldsymbol{a}_i) \boldsymbol{r}_i(k)$$

where $w_{\Phi_k}(x, \boldsymbol{a}) := \pi(\Phi_k(\boldsymbol{a})|x)/\pi_0(\Phi_k(\boldsymbol{a})|x)$ is a generalized importance weight associated with ranking action subset $\Phi_k(\boldsymbol{a}) \in \boldsymbol{a} \in \Pi(\mathcal{A})$ for each position $k$.

**Standard IPS (SIPS) estimator** When $\Phi_k(\boldsymbol{a}) = \boldsymbol{a}$, the GIPS becomes SIPS estimator $\hat{V}_{\text{SIPS}}^{(k)}$. This is the most fundamental estimator in the context of OPE where it is applied. It is unbiased under the following common support assumption. However, it tends to exhibit high variance as the number of ranking actions increases.

**Assumption 2.1. Common Support:** Logging policy $\pi_0$ has common support for policy $\pi$ if $\pi(\boldsymbol{a}|x) > 0 \to \pi_0(\boldsymbol{a}|x) > 0$ for all $\boldsymbol{a} \in \Pi(\mathcal{A})$ and $x \in \mathcal{X}$.

**IIPS Estimator** To address the high variance associated with SIPS, IIPS estimator $\hat{V}_{\text{IIPS}}^{(k)}$ utilizes marginalized importance weights under the assumption of independent user behavior on ranking action spaces (Li et al., 2018) (when $\Phi_k(\boldsymbol{a}) = \boldsymbol{a}(k)$). The IIPS showed low variance owing to its restricted action spaces. Although the IIPS is unbiased if $q_k(x, \boldsymbol{a}) = q_k(x, \boldsymbol{a}(k))$, this assumption is often invalid in practice, leading to significant biases (McInerney et al., 2020).

**RIPS Estimator** To address the high variance of SIPS and the bias of IIPS, RIPS estimator $\hat{V}_{\text{RIPS}}^{(k)}$ utilizes marginalized importance weights under the assumption of cascading

user behavior on ranking action spaces (McInerney et al., 2020) (when $\Phi_k(\boldsymbol{a}) = \boldsymbol{a}(1:k)$). The RIPS is unbiased if $q_k(x, \boldsymbol{a}) = q_k(x, \boldsymbol{a}(1:k))$. This estimator leverages cascading user behavior that aligns with real-world scenarios, where users reward the item in a specific position after being influenced by items positioned above them.

**AIPS Estimator** Although RIPS effectively manages the trade-off between bias and variance in real-world scenarios, it assumes that all users follow the same behavior. To address diverse user behavior, (Kiyohara et al., 2023) developed the AIPS estimator. Notably, we define the AIPS independent of the GIPS.

$$\hat{V}_{\text{AIPS}}^{(k)}(\pi; \mathcal{D}) := \frac{1}{n} \sum_{i=1}^{n} w_{\Phi_k}(x_i, \boldsymbol{a}_i, \boldsymbol{c}_i) \boldsymbol{r}_i(k)$$

where $\boldsymbol{c}$ is a random variable representing user behavior generated from unknown distribution $p(\boldsymbol{c}|x)$ given context $x$, and $\Phi_k(\boldsymbol{a}, \boldsymbol{c})$ is an action subset that influences the reward at position $k$, switching according to the behavior model $\boldsymbol{c}$. For instance, if user behavior variable $\boldsymbol{c}$ follows a cascade behavior, then $\Phi_k(\boldsymbol{a}, \boldsymbol{c}) = \boldsymbol{a}(1:k)$. $w_{\Phi_k}(x, \boldsymbol{a}, \boldsymbol{c}) := \pi(\Phi_k(\boldsymbol{a}, \boldsymbol{c})|x)/\pi_0(\Phi_k(\boldsymbol{a}, \boldsymbol{c})|x)$ is the adaptive importance weight. By assuming that each user adheres to a specific behavior, it becomes feasible to estimate policy value while considering diverse user behaviors.

**Limitation of previous studies** These estimators have been developed to address the high variance and bias associated with increasing the length of rankings. However, because the number of ranking actions depends on both the length of the rankings and the number of unique actions, these estimators may still suffer from high variance as both the number of unique actions and the length of the rankings increase.

## 3. Our Proposed Estimators

Here, we propose a new generalized estimator to address the increased variance associated with both the number of unique actions and length of the ranking. First, to effectively manage the trade-offs between bias and variance, even as the number of unique actions increases, we extend the embedding generation process proposed by (Saito & Joachims, 2022) to ranking settings. We then employ ranking embeddings $\boldsymbol{e} := (\boldsymbol{e}(1), \cdots, \boldsymbol{e}(K)) \in \Pi(\mathcal{E})$, which are generated from $p(\boldsymbol{e}|x, \boldsymbol{a})$ given context $x$ and ranking actions $\boldsymbol{a}$. Here, $\boldsymbol{e}(k) = \boldsymbol{e} \in \mathcal{E}$ represents an action embedding observed at position $k$, whereas $\mathcal{E}$ and $\Pi(\mathcal{E})$ denote the set of unique action and ranking embeddings, respectively. That is, ranking embeddings are derived from a vector of unique action embeddings observed at each position, as illustrated in Figure 1, which assumes that embeddings are observed independently at each position (i.e., $p(\boldsymbol{e}|x, \boldsymbol{a}) = \prod_{k=1}^{K} p(\boldsymbol{e}(k)|x, \boldsymbol{a}(k))$).

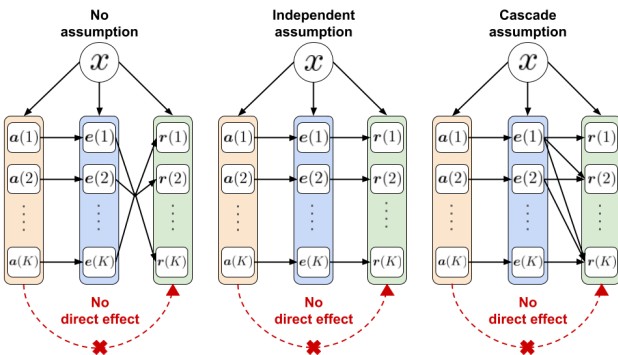

*Figure 1.* Assumption of no direct effect in ranking settings. Ranking actions do not have a causal effect on rewards.

For instance, this is observed when the category (e.g., movie genre) of unique actions for each position is treated as ranking embeddings. We can also assume dependencies, where the embedding at one position is influenced by the actions at other positions [1]. Whether $p(\boldsymbol{e}|x, \boldsymbol{a})$ is known depends on the OPE task. For instance, if a company operating a web service is developing an in-house algorithm in which the price of each action in the ranking varies with $x$, $p(\boldsymbol{e}|x, \boldsymbol{a})$ is likely to be known. Although $\boldsymbol{c}$ in AIPS is unknown, it is left to the developer to decide how to configure $\boldsymbol{e}$.

We then extend the reward function, Eq.(1), and the data generation process, Eq.(2), as follows:

$$V(\pi) := \sum_{k=1}^{K} \underbrace{\mathbb{E}_{p(x)\pi(\boldsymbol{a}|x)p(\boldsymbol{e}|x, \boldsymbol{a})} [q_k(x, \boldsymbol{a}, \boldsymbol{e})]}_{V^{(k)}(\pi)} \quad (3)$$

$$(x, \boldsymbol{a}, \boldsymbol{e}, \boldsymbol{r}) \sim p(x)\pi_0(\boldsymbol{a}|x)p(\boldsymbol{e}|x, \boldsymbol{a})p(\boldsymbol{r}|x, \boldsymbol{a}, \boldsymbol{e}) \quad (4)$$

where $q_k(x, \boldsymbol{a}, \boldsymbol{e}) = \mathbb{E}[\boldsymbol{r}(k)|x, \boldsymbol{a}, \boldsymbol{e}]$ is an expected reward for each position. Since $\mathbb{E}_{\boldsymbol{e}}[q_k(x, \boldsymbol{a}, \boldsymbol{e})] = q_k(x, \boldsymbol{a})$, Eq.(3) becomes an extended formulation of Eq.(1). We can then obtain $n$ logged data $\mathcal{D} = \{(x_i, \boldsymbol{a}_i, \boldsymbol{e}_i, \boldsymbol{r}_i)\}_{i=1}^{n}$ independently and identically from Eq.(4).

**Assumption 3.1.** *Common Ranking Embedding Support:* Logging policy $\pi_0$ is said to have common support for policy $\pi$ if $p(\boldsymbol{e}|x, \pi) > 0 \rightarrow p(\boldsymbol{e}|x, \pi_0) > 0$ for all $\boldsymbol{e} \in \Pi(\mathcal{E})$ and $x \in \mathcal{X}$, where $p(\boldsymbol{e}|x, \pi) := \sum_{\boldsymbol{a} \in \Pi(\mathcal{A})} p(\boldsymbol{e}|x, \boldsymbol{a})\pi(\boldsymbol{a}|x)$ is a marginal distribution over the ranking embedding space given context $x$ and policy $\pi$.

**Assumption 3.2.** *No Direct Effect on Rankings:* Ranking action $\boldsymbol{a}$ have no direct effect on reward $\boldsymbol{r}$, i.e., $\boldsymbol{a} \perp\!\!\!\perp \boldsymbol{r} \mid x, \boldsymbol{e}$.

Even if Assumption 2.1 is not satisfied, while GIPS suffers from bias issues (Sachdeva et al., 2020), the proposed esti-

---

[1]For example, if we decide to set the price of the action at position $k$ based on the price of the action at the position above, the embedding generation process depicted in Figure 1 is merely one example.

mator remains unbiased if Assumptions 3.1 and 3.2 are met. This is one of the primary advantages of marginalization through embeddings. (Saito & Joachims, 2022) discussed this topic exhaustively. We previously stated that whether $p(\boldsymbol{e}|x, \boldsymbol{a})$ is known depends on the OPE task; however, the $p(\boldsymbol{e}|x, \boldsymbol{a})$ required for Assumption 3.2 to hold is clearly unknown.

As shown in Figure 1, under Assumption 3.2, reward $\boldsymbol{r}$ is not influenced by ranking action $\boldsymbol{a}$. Therefore, we only consider the ranking embedding space, which is smaller than the ranking action space. The position-wise effect of our proposed *GMIPS* estimator is as follows:

$$\hat{V}_{\text{GMIPS}}^{(k)}(\pi; \mathcal{D}) := \frac{1}{n} \sum_{i=1}^{n} \underbrace{\frac{p(\Phi_k(\boldsymbol{e}_i)|x_i, \pi)}{p(\Phi_k(\boldsymbol{e}_i)|x_i, \pi_0)}}_{w_{\Phi_k}(x_i, \boldsymbol{e}_i)} \boldsymbol{r}_i(k)$$

where $w_{\Phi_k}(x, \boldsymbol{e})$ is a generalized marginal importance weight over the ranking embedding subset $\Phi_k(\boldsymbol{e}) \in \boldsymbol{e} \in \Pi(\mathcal{E})$.

### 3.1. Marginalized SIPS (MSIPS) Estimator

When $\Phi_k(\boldsymbol{e}) = \boldsymbol{e}$, the GMIPS becomes *the MSIPS* estimator $\hat{V}_{\text{MSIPS}}^{(k)}$. This estimator does not assume specific user behavior on ranking embeddings (on the left of Figure 1). In this case, $w_{\Phi_k}(x, \boldsymbol{e}) := p(\boldsymbol{e}|x, \pi)/p(\boldsymbol{e}|x, \pi_0)$ is the marginal importance weight over ranking embeddings. Although MSIPS can function effectively even with a large number of unique actions if the ranking embedding space is smaller than the ranking action spaces, it may suffer from high variance issues when the ranking is extended owing to its ranking-wise weights.

### 3.2. Doubly Marginalized Estimators

To overcome the high variance issues associated with increasing both the number of unique actions and length of the ranking, we first introduce a new assumption regarding user behavior.

**Assumption 3.3.** *User Behavior Model on Ranking Embedding Spaces:* Reward $\boldsymbol{r}$ is said to follow the specific user behavior model on ranking embedding spaces if $\mathbb{E}_{\boldsymbol{r}}[\boldsymbol{r}|x, \boldsymbol{e}] = \mathbb{E}_{\boldsymbol{r}}[\boldsymbol{r}|x, \Phi_k(\boldsymbol{e})]$ for all $x \in \mathcal{X}, \Phi_k(\boldsymbol{e}) \in \boldsymbol{e} \in \Pi(\mathcal{E})$, and $k \in [K]$.

Assumption 3.3 holds, indicating that the user behavior model based on ranking actions is also applicable to ranking embedding spaces (on the middle and right of Figure 1). Assumption 3.3 states that a necessary and sufficient condition for its validity is that assumption 3.2 holds. Consequently, the following is derived:

**Proposition 3.4.** *If Assumption 3.3 holds, then Assumption 3.2 also holds.*

**Definition 3.5.** *Doubly Marginal Distribution:* The doubly

marginal distribution, based on Assumption 3.3, is defined as follows for all $x \in \mathcal{X}, k \in [K]$, and $\pi$:

$$p(\Phi_k(\boldsymbol{e})|x, \pi) := \sum_{\boldsymbol{e}' \in \Pi(\mathcal{E})} p(\boldsymbol{e}'|x, \pi)\mathbb{I}\{\Phi_k(\boldsymbol{e}) = \Phi_k(\boldsymbol{e}')\}$$

where $\mathbb{I}\{\cdot\}$ is an indicator function that returns 1 if the proposition within it is true and 0 if it is false. To obtain this doubly marginal distribution, we first need to compute the marginal distribution using one-step ranking embeddings $p(\boldsymbol{e}'|x, \pi)$. Definition 3.5 assumes that embeddings are discrete. However, we can also define them in the continuous case by utilizing a kernel function (Kallus & Zhou, 2018).

**MRIPS Estimator** When $\Phi_k(\boldsymbol{e}) = \boldsymbol{e}(1{:}k)$, the GMIPS becomes *the MRIPS* estimator $\hat{V}_{\text{MRIPS}}^{(k)}$. The MRIPS assumes *a cascade model on ranking embeddings* (on the right of Figure 1). In this case, $w_{\Phi_k}(x, \boldsymbol{e}) := p(\boldsymbol{e}(1{:}k)|x, \pi)/p(\boldsymbol{e}(1{:}k)|x, \pi_0)$ is a doubly marginal importance weight by a cascade behavior assumption on ranking embeddings. $p(\boldsymbol{e}(1{:}k)|x, \pi)$ is a doubly marginal distribution as defined in Definition 3.5. To address the variance problem associated with the increased length of rankings, an issue that the MSIPS cannot resolve, the MRIPS employs a cascade assumption on ranking embeddings. This approach provides a more balanced trade-off between bias and variance (McInerney et al., 2020).

**Marginalized IIPS (MIIPS) Estimator** We can also employ an alternative user behavior model in addition to cascade assumption on ranking embeddings. When $\Phi_k(\boldsymbol{e}) = \boldsymbol{e}(k)$, the GMIPS becomes *MIIPS* estimator $\hat{V}_{\text{MIIPS}}^{(k)}$. The MIIPS assumes *an independent model on ranking embeddings* (the middle of Figure 1.). In this case, $w_{\Phi_k}(x, \boldsymbol{e}) := p(\boldsymbol{e}(k)|x, \pi)/p(\boldsymbol{e}(k)|x, \pi_0)$ is a doubly marginal importance weight by an independent behavior assumption on ranking embeddings. $p(\boldsymbol{e}(k)|x, \pi)$ is a doubly marginal distribution as defined in Definition 3.5. Notably, the position-wise effects $\hat{V}_{\text{MIIPS}}^{(k)}$ are equivalent to the MIPS estimator (Saito & Joachims, 2022). See Appendix B for details. However, the MIIPS may introduce a large bias, similar to the conventional IIPS, owing to its simplistic assumption of user behavior, in contrast to the MRIPS.

### 3.3. Theoretical Analysis

Here, we analyze the statistical properties of our estimators. Specifically, we examine the unbiasedness and variance reduction of the GMIPS in comparison to GIPS when Assumption 3.1 and 3.3 are satisfied. We then derive the bias of the GMIPS when Assumption 3.2 is not met and characterize the trade-off between bias and variance in the GMIPS.

**Proposition 3.6.** *Under Assumptions 3.1 and 3.3, the GMIPS is unbiased, i.e., $\mathbb{E}_{p(\mathcal{D})}[\hat{V}_{GMIPS}^{(k)}(\pi; \mathcal{D})] = V^{(k)}(\pi)$ for any policy $\pi$. See Appendix C.1 for the proof.*

Under these assumptions, the GMIPS does not introduce bias while the GIPS does (Sachdeva et al., 2020), even if Assumption 2.1 is not met. The reduction in the variance of the GMIPS in comparison to the GIPS is then as follows:

**Theorem 3.7.** *(Variance Reduction)The GMIPS reduces variance compared to the GIPS, which assumes the same behavior on ranking actions as the GMIPS under Assumptions 2.1, 3.1, and 3.3. See Appendix C.2 for the proof.*

$$n \left( \mathbb{V}_{p(\mathcal{D})} \left[ \hat{V}_{GIPS}^{(k)}(\pi; \mathcal{D}) \right] - \mathbb{V}_{p(\mathcal{D})} \left[ \hat{V}_{GMIPS}^{(k)}(\pi; \mathcal{D}) \right] \right)$$

$$= \mathbb{E}_{x, \Phi_k(e) \sim \pi_0} \left[ \mathbb{E}_r \left[ r(k)^2 \right] \mathbb{V}_{\Phi_k(a)} \left[ w_{\Phi_k}(x, a) \right] \right],$$

The variance of the GMIPS, under these assumptions, is theoretically smaller than that of the GIPS for any given position. Since it contains the variance of $w_{\Phi_k}(x, a)$, and the expectation of $r(k)^2$ given $\Phi_k(e)$, the larger the values, the greater is the reduction in variance. This implies that as the number of ranking actions increases, which is determined by the number of unique actions and the length of the rankings, or as $w_{\Phi_k}(x, a)$ approaches the standard weights, the variance of $w_{\Phi_k}(x, a)$ increases. This substantially reduces variance. However, as this is conditioned on $\Phi_k(e)$, it is important to avoid incorporating excessive information into $e$ to achieve significant variance reduction (Saito & Joachims, 2022).

Although the GMIPS, under several assumptions, can reduce the variance compared to the GIPS, if these assumptions are not met, the GMIPS, specifically MRIPS and MIIPS, may suffer from high bias owing to its double marginalization. Under Assumption 3.1, Proposition 3.4 implies that the GMIPS has two patterns of bias. One is when Assumption 3.2 holds but Assumption 3.3 does not [2]. The other is when even Assumption 3.2 does not hold, which is the most realistic scenario.

**Theorem 3.8.** *(Bias of the GMIPS)Under Assumption 3.1, the GMIPS has the following bias even when Assumption 3.2 does not hold. See Appendix C.4 for proof.*

$$Bias \left( \hat{V}_{GMIPS}^{(k)}(\pi; \mathcal{D}) \right)$$

$$= \mathbb{E}_{x, a, e \sim \pi} \left[ \left( w_{\Phi_k^c}^{-1}(x, e) - 1 \right) q_k(x, a, e) \right]$$

$$+ \mathbb{E}_{x, e \sim \pi_0} \left[ w_{\Phi_k^c}^{-1}(x, e) \sum_{s < t} \pi_0(a_s | x, e) \pi_0(a_t | x, e) \times \right.$$

$$\left. \left( q_k(x, a_s, e) - q_k(x, a_t, e) \right) \left( w(x, a_t) - w(x, a_s) \right) \right],$$

Theorem 3.8 indicates that the bias of the GMIPS is the sum of violations of both Assumptions 3.2 (the second term) and 3.3 (the first term). The first term includes

---

[2]We derived the bias in this case in Appendix C.5.

$w_{\Phi_k^c}^{-1}(x, e) - 1$, which represents the policy deviation of the importance weights for complement set $\Phi_k^c(e)$ of $\Phi_k(e)$, which are not considered owing to their double marginalization, where $w_{\Phi_k^c}(x, e) = \frac{p(\Phi_k^c(e) | x, \pi, \Phi_k(e))}{p(\Phi_k^c(e) | x, \pi_0, \Phi_k(e))}$ is a doubly marginalized weight for $\Phi_k^c(e)$. The second term contains $q_k(x, a_s, e) - q_k(x, a_t, e)$, which is the difference between the expected reward of $a_s$ and $a_t$ given the same $x$ and $e$ when Assumption 3.2 is not satisfied. If Assumption 3.2 holds, the second term becomes zero. Moreover, $w(x, a_t) - w(x, a_s)$ is the difference of the importance weight of $a_s$ and $a_t$, and $\pi_0(a_s | x, e) \pi_0(a_t | x, e)$ is the product of probabilities of selecting $a_s$ and $a_t$ given the same $x$ and $e$. This suggests that as the dimension of $e$ increases, the closer the probability of $\pi(a_s | x, e)$ or $\pi(a_t | x, e)$ approaching deterministic (one approaches zero) values, resulting in the second term approaching zero. That is, we should utilize the high dimensional embeddings, and $w_{\Phi_k}(x, e)$ that are close to standard weights to reduce bias.

**Controlling for GMIPS bias and variance trade-off** From Theorems 3.7 and 3.8 and the fact that the closer $w_{\Phi_k}(x, e)$ is to the independent weight, the greater is the variance reduction for GMIPS, compared to MSIPS(See Appendix C.3), we characterize the trade-off between bias and variance of the GMIPS. To reduce the bias of the GMIPS, we incorporate a substantial amount of information into $e$ and utilize the GMIPS close to MSIPS. In contrast, to reduce the variance of the GMIPS, we avoid incorporating excessive information into $e$, and utilize GMIPS, close to MIIPS. That is, we can only control how to structure the embedding and which of the estimator among the GMIPS variants we use. In the real world, Assumption 3.2 is rarely true. To achieve the minimum MSE in this situation, we can take advantage of data-driven estimator selection methods such as SLOPE (Su et al., 2020b), which is based on Lepski's principle (Lepski & Spokoiny, 1997) that evaluates multiple estimators with different hyper-parameters using the concentration inequality and selects the optimal parameters from logged data. In the following experiments, we show that embedding selection by SLOPE can improve the MSE of the GMIPS.

## 4. Synthetic Experiments

Here, we conduct synthetic experiments to evaluate our proposed estimators across various data environments.

### 4.1. Data Generation Process

We generate synthetic data based on the *Open Bandit Pipeline (OBP)* [3] (Saito et al., 2020), and the synthetic experimental settings of previous studies (Saito & Joachims, 2022; Kiyohara et al., 2022; 2023). See Appendix D

---

[3]https://github.com/st-tech/zr-obp

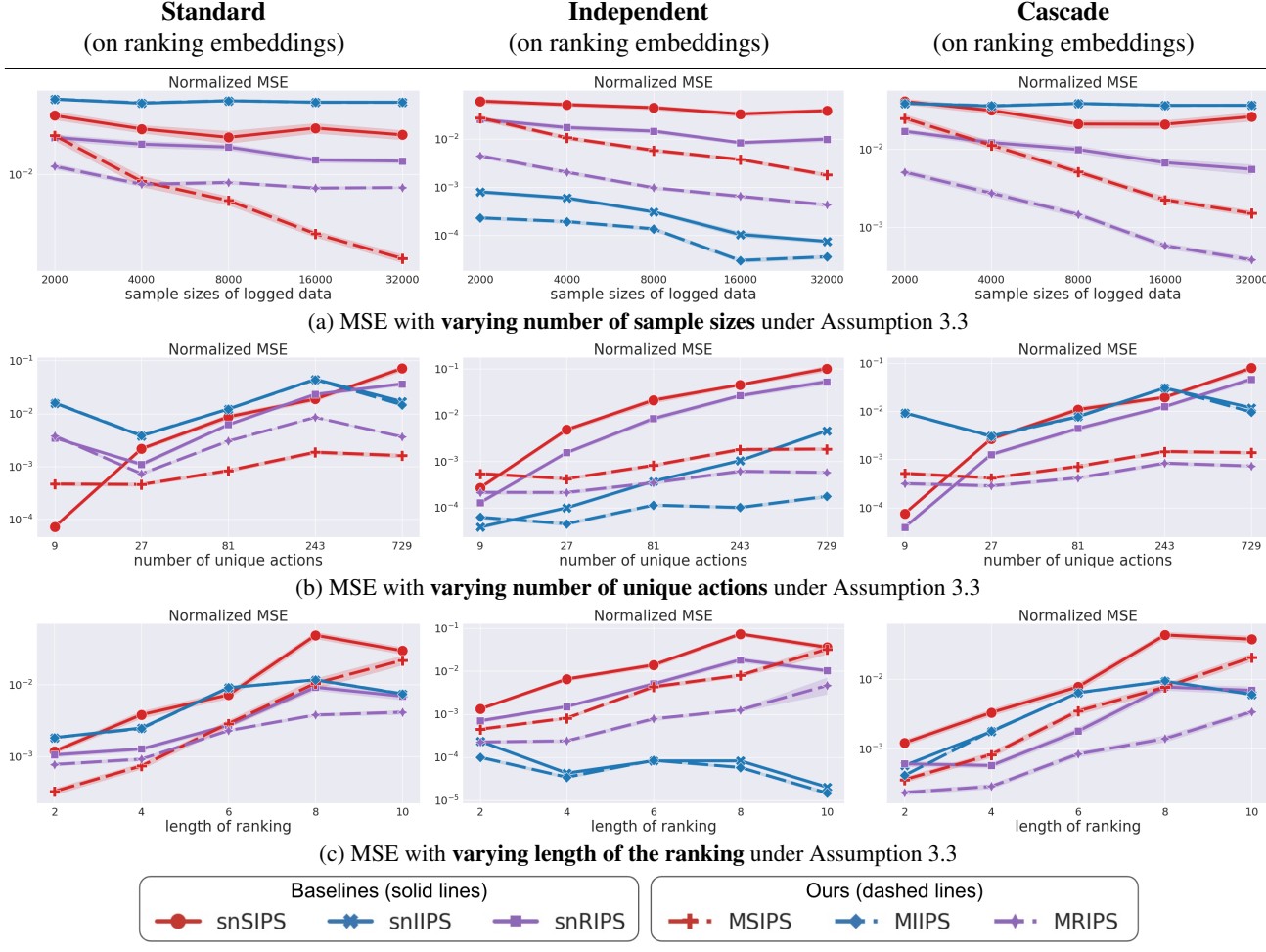

(a) MSE with **varying number of sample sizes** under Assumption 3.3

(b) MSE with **varying number of unique actions** under Assumption 3.3

(c) MSE with **varying length of the ranking** under Assumption 3.3

*Figure 2.* Comparison of MSE normalized by $V(\pi)$ when Assumption 3.3 holds. The solid lines represent existing estimators, whereas the dashed lines indicate our proposed estimators. The colors vary according to the user behavior model assumed for each space. Note that all MSE values are presented on log-scale.

for details. We then obtain $n$ logged data $(x, \boldsymbol{a}, \boldsymbol{e}, \boldsymbol{r}) \sim p(x)\pi_0(\boldsymbol{a}|x)p(\boldsymbol{e}|\boldsymbol{a})p(\boldsymbol{r}|x, \boldsymbol{e})$ independently and identically, where we set $n$ to 10,000, the dimension of $x$ to 5, the number of $|\mathcal{A}_k|$ to 20 for each position, and the length of rankings $K$ to 5 (i.e., $|\mathcal{A}| = |\mathcal{A}_k| \times K = 100$). We then generate binary ranking embeddings and set the number of dimensions to 3 for each position (i.e., $|\Pi(\mathcal{E})| = 2^{3 \times 5} < |\Pi(\mathcal{A})| = 20^5$) as default values. Subsequently, we generated rewards that follow Assumption 3.3 with the independent and cascade assumption in addition to the no(standard) assumption as candidates. Regarding the logging and target policies, we establish an anti-optimal Softmax policy as the logging policy for the expected reward, whereas we set an optimal epsilon-greedy policy as the target policy for the expected reward.

### 4.2. Baselines and Proposed Estimators

**snSIPS, snIIPS, snRIPS**, and **snAIPS (w/UBT)** are conventional estimators that do not utilize ranking embeddings. Owing to the significant high variance problem associated with this limitation, self-normalization (sn) (Swaminathan & Joachims, 2015b) is applied to these estimators. Regarding snAIPS (w/UBT), as the user behavior distribution is always unknown, we optimize the user behavior model using the user behavior tree (UBT) proposed by (Kiyohara et al., 2023). We then define the position-wise effect of snGIPS and snAIPS (w/UBT).

$$\hat{V}_{\text{snGIPS}}^{(k)}(\pi; \mathcal{D}) := \frac{\sum_{i=1}^{n} w_{\Phi_k}(x_i, \boldsymbol{a}_i)\boldsymbol{r}_i(k)}{\sum_{i=1}^{n} w_{\Phi_k}(x_i, \boldsymbol{a}_i)}$$

$$\hat{V}_{\text{snAIPS (w/UBT)}}^{(k)}(\pi; \mathcal{D}) := \frac{\sum_{i=1}^{n} w_{\Phi_k}(x_i, \boldsymbol{a}_i, \hat{\boldsymbol{c}}_i)\boldsymbol{r}_i(k)}{\sum_{i=1}^{n} w_{\Phi_k}(x_i, \boldsymbol{a}_i, \hat{\boldsymbol{c}}_i)}$$

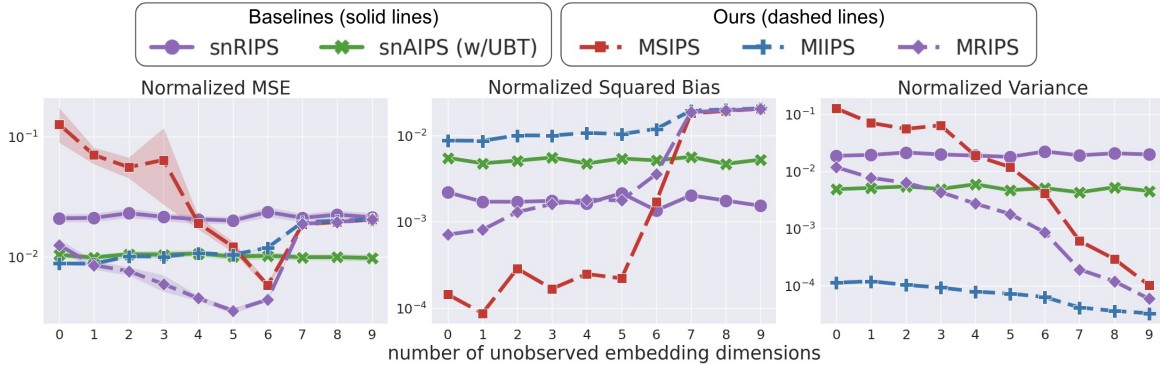

*Figure 3.* MSE, bias, and variance normalized by $V(\pi)$ with **varying number of unobserved dimensions in the ranking embeddings**. Note that these are all log-scale.

where $\hat{c}$ is an estimated user behavior model by UBT for each context $x$.

**MSIPS**,**MIIPS**,**MRIPS** are our proposed estimators that utilize ranking embeddings to address the limitations of the aforementioned estimators.

### 4.3. Results

This section presents the MSE, squared bias, and variance (all normalized by $V(\pi)$) for the proposed estimators. These metrics were computed 1000 times using sets of logged data, each replicated with different random seeds.

**Performance of our proposed estimators with varying sample sizes** We varied the sample sizes $\{2000, 4000, 8000, 16000, 32000\}$ in the logged data. The results are presented in Figure 2(a). We observed that our unbiased estimators, as shown in each figure, outperformed the existing estimators as the sample size increases. Specifically, the performance of the MSIPS, MIIPS, and MRIPS improved by approximately 92.5%, 93.0%, and 51.9%, respectively, as compared to that of the snSIPS, snIIPS, and snRIPS, in the order presented in the figure (left to right) when the sample size was 32000.

**Performance of our proposed estimators with varying number of unique actions** We set $K = 3$ and varied the number of unique actions at position $|\mathcal{A}_k| \in \{3, 9, 27, 81, 243\}$ (i.e., $|\mathcal{A}| = |\mathcal{A}_k| \times K$). The results are presented in Figure 2(b). We observed that our unbiased estimators outperformed the existing estimators as the unique actions increases. Specifically, the performance of the MSIPS, MIIPS, and MRIPS improved by approximately 97.8%, 98.4%, and 96.2%, respectively, as compared to that of the snSIPS, snIIPS, and snRIPS, in the order presented in the figure (left to right) when $|\mathcal{A}| = 729$. We achieved a significant MSE reduction compared to snSIPS and snRIPS, which exhibited particularly high variances phenomenon, which can be explained by Theorem 3.7.

**Performance of our proposed estimators with varying length of the ranking** We set $|\mathcal{A}_k| = 8, \forall k \in [K]$ and varied the length of the ranking $K \in \{2, 4, 6, 8, 10\}$. The results are presented in Figure 2(c). We observed that our unbiased estimators, other than MSIPS, outperformed the existing estimators even as the length of the rankings increases. In contrast, the performance of the MSIPS deteriorated owing to its ranking-wise weight. Specifically, the performance of MSIPS, MIIPS, and MRIPS improved by approximately 26.4%, 81.0%, and 27.4%, respectively, as compared to that of the snSIPS, snIIPS, and snRIPS, in the order presented in the figure (left to right) when the length of the rankings is 10. This suggests that the MRIPS, which leverages double marginalization based on a cascade model on ranking embeddings, remains effective even as the length of the rankings increases.

**Performance of our proposed estimators if Assumption 3.2. does not hold** We set $D = 10$ and varied the number of unobserved embedding dimensions from 0 to 9. This indicates that as the number of unobserved dimensions approaches 9, the degree of violation of Assumption 3.2 increases. The results are presented in Figure 3. We observed that by intentionally not using a specific dimension, the MSE of MRIPS and MSIPS were lower than those of snRIPS and snAIPS (w/UBT), whose MSE remains constant along the x-axis. Specifically, MRIPS improves by approximately 64.6% compared to snAIPS (w/UBT), and is superior to any other estimators, although snAIPS(w/UBT) optimizes the behavior model for each context when the number of unobserved dimensions is five. Focusing on bias and variance, these trade-offs are influenced by the extent to which both Assumptions 3.2 and 3.3 are violated. Specifically, as the number of unobserved dimensions increases and the GMIPS approaches the MIIPS, which employs independent weights, the bias increases. This observation can be explained by Theorem 3.8. Conversely, as the number of unobserved dimensions increases, the variance of the GMIPS decreases.

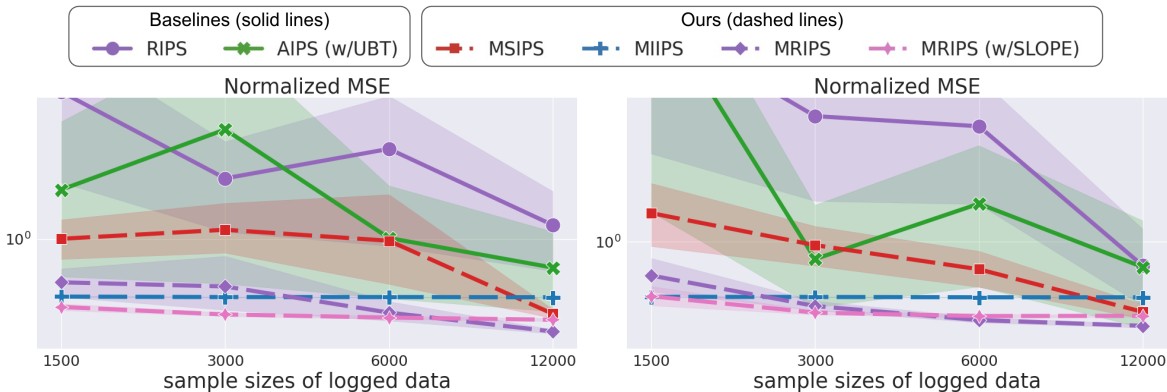

*Figure 4.* MSE normalized by $V(\pi)$ with **varying sample sizes on EUR-Lex4K (left) and RCV1-2K (right) datasets**. Note that all MSE values are presented on log-scale. The computations are performed 1000 times using logged data replicated with different random seeds.

Notably, the MRIPS exhibits a more significant reduction in variance compared to the snRIPS, which can also be clarified by Theorem 3.7.

## 5. Real-World Data Experiments

To assess our proposed estimators in read-world scenarios, where the ranking embeddings $e$ necessary for Assumption 3.2 to hold are unknown, we transformed the EUR-Lex4K and RCV1-2K datasets, comprising pairs of continuous features and the large number of labels from the Extreme Classification Repository (Bhatia et al., 2016) into bandit settings as outlined in (Su et al., 2020a; Saito et al., 2023; Kiyohara et al., 2024).

**Setup** First, we applied PCA (Abdi & Williams, 2010) to these features and subsequently created 10-dimensional contexts $x$. We then established the base expected reward, $\bar{q}(x, a) = 1 - \eta_a$, if the label is positive, and $\bar{q}(x, a) = \eta_a - 1$ otherwise, where $\eta_a$ is a reward noise sampled from a uniform distribution within the range $[0, 0.2]$ for each $a$. Thereafter, we defined expected rewards $q_k(x, \boldsymbol{a}, \boldsymbol{c})$ that reflect diverse user behavior using $\bar{q}(x, a)$ (see Appendix D for details), where we randomly sampled $|\mathcal{A}| = 100$ from all labels and set $|\mathcal{A}_k| = 20$ and $K = 5$ (i.e., $|\mathcal{A}| = |\mathcal{A}_k| \times K$). Since these datasets do not contain action embeddings, we trained an abstraction model in which the middle layer serves as the action embedding within a three-layer neural network. This model generates an action embedding $e_a$ that corresponds to each $a$ based on $x$ and $a$ in advance. We set the dimension of the action embeddings to 15. Subsequently, we discretized the embedding for each dimension, categorizing them into 3. Notably, we obtained deterministic action embeddings. Finally, we observed a binary reward $\boldsymbol{r}(k)$ for each $k$. Regarding the logging and target policies, we applied a Softmax policy to an anti-optimal estimated base reward function $-\hat{\bar{q}}_k(x, \boldsymbol{a}(k))$ that we use ridge regression for each $k$ as the logging policy while employing

an epsilon-greedy policy to an optimal $\hat{\bar{q}}_k(x, \boldsymbol{a}(k))$ as the target policy.

**Results** The results present the MSE normalized by $V(\pi)$ for the estimators discussed in the previous section, along with MRIPS (w/SLOPE), which selects the optimal number of unobserved dimensions using the SLOPE algorithm (Su et al., 2020b) based solely on logged data(See Appendix E for details). We varied the sample sizes $\{1500, 3000, 6000, 12000\}$ in the logged data. The results are presented in Figure 4. We observed that our estimators exhibited the lowest MSE compared to AIPS (w/UBT) and RIPS, which suffer from high variance issues owing to their importance weights on ranking actions, although AIPS (w/UBT) optimizes user behavior model for each context. Notably, by selecting the optimal embedding dimensions, the MRIPS (w/SLOPE) achieved the lowest MSE other than estimators, and improved in MSE approximately 73.7%(left) and 96.5% (right), compared to AIPS (w/UBT) when sample size was 1500. As the data size increases, the MIIPS maintains a constant MSE, whereas the MRIPS and MSIPS approach optimal performance.

## 6. Conclusion

We proposed the use of embeddings and GMIPS to address the variance problem associated with existing estimators as the number of unique actions and length of ranking increase. Through theoretical analysis and experiments, we demonstrated that, among the GMIPS, the MRIPS, which assumes a cascade behavior model on ranking embeddings, effectively balances the trade-off between bias and variance. Furthermore, it performs optimally when the appropriate embedding is selected. However, GMIPS may encounter the bias-variance dilemma if a higher-dimensional embedding is necessary to satisfy Assumption 3.2. To address this issue, a ranking extension of the conjunct effect model (Saito et al., 2023) may be effective.

## Impact Statement

This paper presents research aimed at advancing the field of Machine Learning. While there are numerous potential societal implications of our work, we believe that none require specific emphasis in this context.

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

# A. Related Works

Here, we provide an overview of the related works pertinent to our study.

## A.1. Off-Policy Evaluation (OPE)

As mentioned in the main text, OPE aims to accurately estimate the policy value of the target policy to make decisions using only logged bandit data collected from previous versions.

**OPE in single-action decision making** Single-action decision-making, such as selecting medications and advertising strategies, is the simplest task among the OPE. The basic estimators in this context include the direct method (DM), inverse propensity score (IPS), and doubly robust (DR) estimators (Dudík et al., 2011; 2014). DM utilizes a directly estimated reward function in advance. However, although DM exhibits low variance, it suffers from significant bias owing to estimation errors in the reward function. IPS employs an importance weight, which reflects the divergence between target and logging policies to estimate policy value. As IPS accounts for the propensity of logging policies, it is unbiased. However, IPS generates large variance owing to discrepancies between target and logging policies. To address the limitations of both DM and IPS, DR incorporates the importance weight and estimated reward function. Among these estimators, DR can achieve the minimum MSE. Other methods have been proposed to reduce the variance of IPS by permitting a certain level of bias through self-normalization and the clipping of importance weights (Swaminathan & Joachims, 2015b). However, as the action spaces rises, the performance of these estimators, including DR, deteriorates owing to variance issues related to large action spaces. To mitigate this problem, the marginalized IPS (MIPS) and off-policy estimator via conjunct effect modeling (OffCEM) estimators leverage action embeddings or clustering techniques to minimize the impact of large action spaces, thereby reducing the variance of the estimators (Saito & Joachims, 2022; Saito et al., 2023). If the assumption of no direct effect holds, meaning that only the context and action embeddings influence the reward function, MIPS becomes unbiased. Furthermore, if the dimensionality of the action embeddings is smaller than that of the action space, the variance reduction of the MIPS increases compared to that of the IPS, resulting in the MIPS achieving a lower MSE than the IPS. However, to satisfy the assumption of no direct effect, high dimensional action embeddings are required, which can lead to bias-variance dilemma (i.e., the dimensionality and quality of action embeddings governs the balance between bias and variance ). To overcome the limitations of the MIPS, OFFCEM leverages not only the marginalized importance weight over the cluster space but also the estimated reward function, which is derived from two-step regressions. The name "CEM" is derived from the assumption that the reward function can be decomposed into cluster and residual effects, which the MIPS does not consider. The structure of the OFFCEM is similar to that of the DR. However, it differs in that the importance weights and the estimated reward function serve distinct roles. Empirical evidence suggests that if the assumption of local correctness holds, the OFFCEM becomes unbiased and exhibits variance reduction compared to all other OPE estimators. Furthermore, in practice, on music recommendation platforms, the estimator that utilizes the structure of CEM outperforms the typical ope estimators, such as IPS (Saito et al., 2024a).

**OPE in multiple-action decision making** For multiple-action decision making, such as ranking and slate recommendations, the distinction between ranking and slate lies in the observation of rewards. In ranking settings, such as video recommendations, we select ranking actions where a unique action is chosen for each position. This enables us to obtain a vector of rewards (e.g., watch time) for each position. In contrast, slate settings, such as thumbnail recommendations, involve selecting the picture, color of the thumbnail, and title text simultaneously, resulting in a scalar reward (e.g., watch time). Research in these areas is still in its infancy, particularly concerning large action spaces (Kiyohara et al., 2024; Shimizu et al., 2024). The complexity increases owing to the exponential growth of candidate ranking (slate) actions compared to single-action decision-making. Hence, as the DM and DR, which estimate all rewards, are not suitable for this context (McInerney et al., 2020), numerous studies have introduced various assumptions into the reward function and developed variants of IPS that are more appropriate (Li et al., 2018; McInerney et al., 2020; Kiyohara et al., 2023; Swaminathan et al., 2017; Kiyohara et al., 2024). Our study in one such example. As explained in the main text, (Li et al., 2018; McInerney et al., 2020; Kiyohara et al., 2023) addressed the variance problem caused by the length of the ranking by assuming user behavior regarding the rewards obtained as a vector. Regarding slate, (Swaminathan et al., 2017) demonstrated that the pseudo inverse (PI) estimator, as the sum of importance weights per slot, is valid under the assumption of linearity in rewards. However, if this assumption is not met, the PI exhibits significant bias (Kiyohara et al., 2024). To address the bias of the PI and the variance introduced by the large slate space, (Kiyohara et al., 2024) developed the latent IPS (LIPS) estimator. The LIPS effectively manages the trade-off by learning the optimal representation from logged data and leveraging the latent importance weights. (Shimizu et al., 2024) addressed the large action space by developing an estimator, off-policy estimator for combinatorial bandits (OPCB), which leverages the structure of OFFCEM for OPE in scenarios with multiple action

choices.

**Estimator selection for OPE** In OPE, MSE is commonly used to identify the most effective estimators, regardless of whether they include hyperparameters (Su et al., 2020a;b; Tucker & Lee, 2021) or stem from different estimator classes (Udagawa et al., 2023). However, selecting estimators using only logged data is challenging, as their biases are influenced by the true policy value. Regarding the tuning of estimators with hyperparameters, MSE surrogates have been proposed, which utilize unbiased estimators, such as IPS and DR (Su et al., 2020a). In contrast, evaluation methods based on probability inequalities avoid estimation through surrogates (Su et al., 2020b; Tucker & Lee, 2021). Although the MSE surrogates are easy to implement, estimating the MSE remains challenging because the performance of the unbiased estimator highly depends on the estimation of bias (Su et al., 2020b; Udagawa et al., 2023). To avoid estimation of bias, (Su et al., 2020b; Tucker & Lee, 2021) proposed selection by Lepski's principle for off-policy evaluation (SLOPE) procedure, which is based on Lepski's principle (Lepski & Spokoiny, 1997) that evaluates multiple estimators with different hyperparameters using the concentration inequality and selects the optimal parameters from logged data. Regarding the selection among various classes of estimators (e.g., whether IPS or DR is better), (Udagawa et al., 2023) proposed the policy adaptive estimator selection via importance fitting (PAS-IF). This method facilitates the selection of different estimators by learning a pseudo-policy, which enables the pseudo-online experimental performance of a target policy based on samples taken from logged data. In relation to our study, we employed the SLOPE procedure to determine the optimal embedding dimension for use in GMIPS. Additionally, it is possible to select between MSIPS and MRIPS with SLOPE through PAS-IF.

### A.2. Off-Policy Learning (OPL)

OPL aims to find the optimal target policy using only logged data collected from previous versions. There are two primary types of learning methods in OPL: regression- and policy gradient-based methods (Jeunen et al., 2020). The regression-based approach involves training a model that directly predicts the expected reward, subsequently determining which action to take based on these predictions. However, regression-based methods are susceptible to bias owing to prediction errors (Saito et al., 2024b). In contrast, policy gradient-based methods utilize the policy gradients of the IPS and DR estimators to learn models that directly predict the probability of action selection, similar to multi-class classification problems (Swaminathan & Joachims, 2015a;b). A significant challenge with these methods is that the variance of the policy gradients tends to increase as the number of possible actions rises (Saito et al., 2024b). To address these bias–variance trade-offs, (Saito et al., 2024b) proposed the policy optimization via two-stage policy decomposition (POTEC) procedure, which combines regression- and policy gradient-based learning. This method introduces a two-step process in which cluster selection is learned in a policy gradient-based manner, whereas the action within the clusters is obtained from a regression-based approach. There are also policy gradient-based OPL studies in the ranking setting. In particular, ranking strategies based on the Plackett–Luce model (Plackett, 1975) are often developed to prevent the duplication of unique actions across positions. Additionally, a policy learning method that considers fairness among items has been proposed to enable a greater number of items to engage the user compared to single-action decisions (Singh & Joachims, 2019; Oosterhuis, 2021).

## B. Example of User behavior Model on Ranking Embeddings (Assumption 3.3)

*Table 1.* Toy examples of user behavior models on ranking embeddings. If the no (standard) or cascade model on ranking embeddings is valid, Assumption 3.2 does not necessarily hold between positions (left). Conversely, if the independent model on ranking embeddings is valid, Assumption 3.2 holds true between positions (right). Specifically, the right side of Table 1 shows that actions with embeddings $e(1) = e(2) = 1$ are selected for positions $k=1, 2$, and the assumption of no direct effect is satisfied between these positions (i.e., $q_1(x, \cdot, e) = q_2(x, \cdot, e) = 0.8$). In contrast, as shown on the left side of the table, the reward for position $k$ is influenced by the embedding of position $k$-1 and is determined even when the embeddings of positions $k$ and $k$-1 are identical. Consequently, the assumption of no direct effect does not hold between these positions (i.e., $q_1(x, \cdot, e) = 0.8, q_2(x, \cdot, e) = 0.5$).

**No assumption** or **Cascade model**
(on ranking embeddings)

| $k$ | $a_1$ | $e$ | $q_k(x, a_1, e)$ | $a_2$ | $e$ | $q_k(x, a_2, e)$ |
|---|---|---|---|---|---|---|
| 1 | 4 | 1 | 0.8 | 8 | 1 | 0.8 |
| 2 | 3 | 1 | 0.5 | 7 | 1 | 0.5 |
| 3 | 2 | 2 | 0.6 | 6 | 2 | 0.6 |
| 4 | 1 | 2 | 0.2 | 5 | 2 | 0.2 |

**Independent model**
(on ranking embeddings)

| $k$ | $a_1$ | $e$ | $q_k(x, a_1, e)$ | $a_2$ | $e$ | $q_k(x, a_2, e)$ |
|---|---|---|---|---|---|---|
| 1 | 4 | 1 | 0.8 | 8 | 1 | 0.8 |
| 2 | 3 | 1 | 0.8 | 7 | 1 | 0.8 |
| 3 | 2 | 2 | 0.9 | 6 | 2 | 0.9 |
| 4 | 1 | 2 | 0.9 | 5 | 2 | 0.9 |

## C. Proofs, Additional Theorems, and Analysis

Here, we present proofs of the proposed theorems that were omitted from the main text along with an analysis of additional theorems.

### C.1. Proof of Proposition 3.6.

*Proof.*

$$\mathbb{E}_{p(\mathcal{D})}\left[\hat{V}_{\text{GMIPS}}(\pi; \mathcal{D})\right]$$

$$= \frac{1}{n}\sum_{i=1}^{n}\mathbb{E}_{p(x_i)\pi_0(\boldsymbol{a}_i|x_i)p(\boldsymbol{e}_i|x_i,\boldsymbol{a}_i)p(\boldsymbol{r}_i|x_i,\boldsymbol{a}_i,\boldsymbol{e}_i)}\left[w_{\Phi_k}(x_i,\boldsymbol{e}_i)\boldsymbol{r}_i(k)\right]$$

$$= \mathbb{E}_{p(x)\pi_0(\boldsymbol{a}|x)p(\boldsymbol{e}|x,\boldsymbol{a})p(\boldsymbol{r}|x,\boldsymbol{a},\boldsymbol{e})}\left[w_{\Phi_k}(x,\boldsymbol{e})\boldsymbol{r}(k)\right]$$

$$= \mathbb{E}_{p(x)}\left[\sum_{\boldsymbol{a}\in\Pi(\mathcal{A})}\pi_0(\boldsymbol{a}|x)\sum_{\boldsymbol{e}\in\Pi(\mathcal{E})}p(\boldsymbol{e}|x,\boldsymbol{a})w_{\Phi_k}(x,\boldsymbol{e})q_k(x,\boldsymbol{a},\boldsymbol{e})\right]$$

$$= \mathbb{E}_{p(x)}\left[\sum_{\boldsymbol{a}\in\Pi(\mathcal{A})}\pi_0(\boldsymbol{a}|x)\sum_{\boldsymbol{e}\in\Pi(\mathcal{E})}p(\boldsymbol{e}|x,\boldsymbol{a})w_{\Phi_k}(x,\boldsymbol{e})q_k(x,\Phi_k(\boldsymbol{e}))\right] \qquad \because \text{Assumption 3.3}$$

$$= \mathbb{E}_{p(x)}\left[\sum_{\boldsymbol{e}\in\Pi(\mathcal{E})}w_{\Phi_k}(x,\boldsymbol{e})q_k(x,\Phi_k(\boldsymbol{e}))\sum_{\boldsymbol{a}\in\Pi(\mathcal{A})}\pi_0(\boldsymbol{a}|x)p(\boldsymbol{e}|x,\boldsymbol{a})\right]$$

$$= \mathbb{E}_{p(x)}\left[\sum_{\boldsymbol{e}\in\Pi(\mathcal{E})}p(\boldsymbol{e}|x,\pi_0)w_{\Phi_k}(x,\boldsymbol{e})q_k(x,\Phi_k(\boldsymbol{e}))\right] \qquad \because p(\boldsymbol{e}|x,\pi_0)=\sum_{\boldsymbol{a}\in\Pi(\mathcal{A})}\pi_0(\boldsymbol{a}|x)p(\boldsymbol{e}|x,\boldsymbol{a})$$

$$= \mathbb{E}_{p(x)}\left[\sum_{\boldsymbol{e}\in\Pi(\mathcal{E})}p(\Phi_k(\boldsymbol{e})|x,\pi_0)p(\Phi_k^c(\boldsymbol{e})|x,\pi_0,\Phi_k(\boldsymbol{e}))w_{\Phi_k}(x,\boldsymbol{e})q_k(x,\Phi_k(\boldsymbol{e}))\right] \qquad (5)$$

$$= \mathbb{E}_{p(x)}\left[\sum_{\boldsymbol{e}\in\Pi(\mathcal{E})}p(\Phi_k(\boldsymbol{e})|x,\pi)p(\Phi_k^c(\boldsymbol{e})|x,\pi_0,\Phi_k(\boldsymbol{e}))q_k(x,\Phi_k(\boldsymbol{e}))\right]$$

$$= \mathbb{E}_{p(x)}\left[\sum_{\boldsymbol{e}\in\Pi(\mathcal{E})}\sum_{\boldsymbol{e}'\in\Pi(\mathcal{E})}\Big(p(\boldsymbol{e}'|x,\pi)\mathbb{I}\{\Phi_k(\boldsymbol{e})=\Phi_k(\boldsymbol{e}')\}\Big)p(\Phi_k^c(\boldsymbol{e})|x,\pi_0,\Phi_k(\boldsymbol{e}))q_k(x,\Phi_k(\boldsymbol{e}))\right] \qquad \because \text{Definition 3.5}$$

$$= \mathbb{E}_{p(x)}\left[\sum_{\boldsymbol{e}'\in\Pi(\mathcal{E})}p(\boldsymbol{e}'|x,\pi)q_k(x,\Phi_k(\boldsymbol{e}'))\underbrace{\sum_{\boldsymbol{e}\in\Pi(\mathcal{E})}p(\Phi_k^c(\boldsymbol{e})|x,\pi_0,\Phi_k(\boldsymbol{e}))\mathbb{I}\{\Phi_k(\boldsymbol{e})=\Phi_k(\boldsymbol{e}')\}}_{=1}\right] \qquad (6)$$

$$= \mathbb{E}_{p(x)}\left[\sum_{\boldsymbol{a}\in\Pi(\mathcal{A})}\pi(\boldsymbol{a}|x)\sum_{\boldsymbol{e}'\in\Pi(\mathcal{E})}p(\boldsymbol{e}'|x,\boldsymbol{a})q_k(x,\boldsymbol{a},\boldsymbol{e}')\right] \qquad \because \text{Assumption 3.3}$$

$$= V^{(k)}(\pi)$$

where we use $p(\boldsymbol{e}|x,\pi_0)=p(\Phi_k(\boldsymbol{e})|x,\pi_0)p(\Phi_k^c(\boldsymbol{e})|x,\pi_0,\Phi_k(\boldsymbol{e}))$ in Eq.(5). Here, if $\Phi_k(\boldsymbol{e})=\boldsymbol{e}$, $\Phi_k^c(\boldsymbol{e})$ becomes $\emptyset$. In this case, we assume $p(\emptyset|x,\pi_0,\boldsymbol{e})=1$ for any $x$, $\pi_0$, and $\boldsymbol{e}$. Regarding Eq.(6), as $p(\Phi_k^c(\boldsymbol{e})|x,\pi_0,\Phi_k(\boldsymbol{e}))\mathbb{I}\{\Phi_k(\boldsymbol{e})=\Phi_k(\boldsymbol{e}')\}=p(\Phi_k^c(\boldsymbol{e})|x,\pi_0,\Phi_k(\boldsymbol{e}'))$ is a conditional distribution given $x$, $\pi_0$, and $\Phi_k(\boldsymbol{e}')$, the sum of this distribution is one.

□

## C.2. Proof of Theorem 3.7.

*Proof.* Under Assumptions 2.1, 3.1, and 3.3, the GIPS and GMIPS estimators are both unbiased. The difference in their variance is then attributed to their second moment, which is calculated as follows.

$$n\left(\mathbb{V}_{p(\mathcal{D})}\left[\hat{V}_{\text{GIPS}}(\pi;\mathcal{D})\right] - \mathbb{V}_{p(\mathcal{D})}\left[\hat{V}_{\text{GMIPS}}(\pi;\mathcal{D})\right]\right)$$

$$= n\left(\mathbb{V}_{p(\mathcal{D})}\left[w_{\Phi_k}(x,\boldsymbol{a})\boldsymbol{r}(k)\right] - \mathbb{V}_{p(\mathcal{D})}\left[w_{\Phi_k}(x,\boldsymbol{e})\boldsymbol{r}(k)\right]\right)$$

$$= \mathbb{E}_{p(x)\pi_0(\boldsymbol{a}|x)p(\boldsymbol{e}|x,\boldsymbol{a})p(\boldsymbol{r}|x,\boldsymbol{a},\boldsymbol{e})}\left[w_{\Phi_k}^2(x,\boldsymbol{a})\boldsymbol{r}(k)^2\right] - \mathbb{E}_{p(x)\pi_0(\boldsymbol{a}|x)p(\boldsymbol{e}|x,\boldsymbol{a})p(\boldsymbol{r}|x,\boldsymbol{a},\boldsymbol{e})}\left[w_{\Phi_k}^2(x,\boldsymbol{e})\boldsymbol{r}(k)^2\right]$$

$$= \mathbb{E}_{p(x)\pi_0(\boldsymbol{a}|x)}\left[\sum_{\boldsymbol{e}\in\Pi(\mathcal{E})} p(\boldsymbol{e}|x,\boldsymbol{a})\left(w_{\Phi_k}^2(x,\boldsymbol{a}) - w_{\Phi_k}^2(x,\boldsymbol{e})\right)\mathbb{E}_{p(\boldsymbol{r}|x,\Phi_k(\boldsymbol{e}))}[\boldsymbol{r}(k)^2]\right] \quad \because \text{Assumption 3.3}$$

$$= \mathbb{E}_{p(x)\pi_0(\boldsymbol{a}|x)}\left[\sum_{\Phi_k(\boldsymbol{e})} \left(w_{\Phi_k}^2(x,\boldsymbol{a}) - w_{\Phi_k}^2(x,\boldsymbol{e})\right)\mathbb{E}_{p(\boldsymbol{r}|x,\Phi_k(\boldsymbol{e}))}[\boldsymbol{r}(k)^2] \sum_{\Phi_k^c(\boldsymbol{e})} p(\Phi_k(\boldsymbol{e}),\Phi_k^c(\boldsymbol{e})|x,\boldsymbol{a})\right] \tag{7}$$

$$= \mathbb{E}_{p(x)}\left[\sum_{\boldsymbol{a}\in\Pi(\mathcal{A})} \pi_0(\boldsymbol{a}|x)\sum_{\Phi_k(\boldsymbol{e})} p(\Phi_k(\boldsymbol{e})|x,\boldsymbol{a})\left(w_{\Phi_k}^2(x,\boldsymbol{a}) - w_{\Phi_k}^2(x,\boldsymbol{e})\right)\mathbb{E}_{p(\boldsymbol{r}|x,\Phi_k(\boldsymbol{e}))}[\boldsymbol{r}(k)^2]\right]$$

$$\because p(\Phi_k(\boldsymbol{e})|x,\boldsymbol{a}) = \sum_{\Phi_k^c(\boldsymbol{e})} p(\Phi_k(\boldsymbol{e}),\Phi_k^c(\boldsymbol{e})|x,\boldsymbol{a})$$

$$= \mathbb{E}_{p(x)}\left[\sum_{\Phi_k(\boldsymbol{a})}\sum_{\Phi_k^c(\boldsymbol{a})} \pi_0(\Phi_k(\boldsymbol{a})|x)\pi_0(\Phi_k^c(\boldsymbol{a})|x,\Phi_k(\boldsymbol{a}))\right.$$

$$\left.\sum_{\Phi_k(\boldsymbol{e})} p(\Phi_k(\boldsymbol{e})|x,\Phi_k(\boldsymbol{a}),\Phi_k^c(\boldsymbol{a}))\left(w_{\Phi_k}^2(x,\boldsymbol{a}) - w_{\Phi_k}^2(x,\boldsymbol{e})\right)\mathbb{E}_{p(\boldsymbol{r}|x,\Phi_k(\boldsymbol{e}))}\left[\boldsymbol{r}(k)^2\right]\right] \tag{8}$$

$$= \mathbb{E}_{p(x)}\left[\sum_{\Phi_k(\boldsymbol{a})} \pi_0(\Phi_k(\boldsymbol{a})|x)\sum_{\Phi_k(\boldsymbol{e})} \left(w_{\Phi_k}^2(x,\boldsymbol{a}) - w_{\Phi_k}^2(x,\boldsymbol{e})\right)\mathbb{E}_{p(\boldsymbol{r}|x,\Phi_k(\boldsymbol{e}))}\left[\boldsymbol{r}(k)^2\right]\right.$$

$$\left.\sum_{\Phi_k^c(\boldsymbol{a})} \pi_0(\Phi_k^c(\boldsymbol{a})|x,\Phi_k(\boldsymbol{a}))p(\Phi_k(\boldsymbol{e})|x,\Phi_k(\boldsymbol{a}),\Phi_k^c(\boldsymbol{a}))\right]$$

$$= \mathbb{E}_{p(x)}\left[\sum_{\Phi_k(\boldsymbol{a})} \pi_0(\Phi_k(\boldsymbol{a})|x)\sum_{\Phi_k(\boldsymbol{e})} p(\Phi_k(\boldsymbol{e})|x,\Phi_k(\boldsymbol{a}))\left(w_{\Phi_k}^2(x,\boldsymbol{a}) - w_{\Phi_k}^2(x,\boldsymbol{e})\right)\mathbb{E}_{p(\boldsymbol{r}|x,\Phi_k(\boldsymbol{e}))}\left[\boldsymbol{r}(k)^2\right]\right]$$

$$\because p(\Phi_k(\boldsymbol{e})|x,\Phi_k(\boldsymbol{a})) = \sum_{\Phi_k^c(\boldsymbol{a})} \pi_0(\Phi_k^c(\boldsymbol{a})|x,\Phi_k(\boldsymbol{a}))p(\Phi_k(\boldsymbol{e})|x,\Phi_k(\boldsymbol{a}),\Phi_k^c(\boldsymbol{a}))$$

$$= \mathbb{E}_{p(x)}\left[\sum_{\Phi_k(\boldsymbol{a})} \pi_0(\Phi_k(\boldsymbol{a})|x)\sum_{\Phi_k(\boldsymbol{e})} \frac{p(\Phi_k(\boldsymbol{e})|x,\pi_0)\pi_0(\Phi_k(\boldsymbol{a})|x,\Phi_k(\boldsymbol{e}))}{\pi_0(\Phi_k(\boldsymbol{a})|x)}\left(w_{\Phi_k}^2(x,\boldsymbol{a}) - w_{\Phi_k}^2(x,\boldsymbol{e})\right)\mathbb{E}_{p(\boldsymbol{r}|x,\Phi_k(\boldsymbol{e}))}\left[\boldsymbol{r}(k)^2\right]\right]$$

$$\because p(\Phi_k(\boldsymbol{e})|x,\Phi_k(\boldsymbol{a})) = \frac{p(\Phi_k(\boldsymbol{e})|x,\pi_0)\pi_0(\Phi_k(\boldsymbol{a})|x,\Phi_k(\boldsymbol{e}))}{\pi_0(\Phi_k(\boldsymbol{a})|x)}$$

$$= \mathbb{E}_{p(x)}\left[\sum_{\Phi_k(\boldsymbol{e})} p(\Phi_k(\boldsymbol{e})|x,\pi_0)\mathbb{E}_{p(\boldsymbol{r}|x,\Phi_k(\boldsymbol{e}))}\left[\boldsymbol{r}(k)^2\right]\underbrace{\sum_{\Phi_k(\boldsymbol{a})} \pi_0(\Phi_k(\boldsymbol{a})|x,\Phi_k(\boldsymbol{e}))\left(w_{\Phi_k}^2(x,\boldsymbol{a}) - w_{\Phi_k}^2(x,\boldsymbol{e})\right)}_{\text{①}}\right] \tag{9}$$

$$= \mathbb{E}_{p(x)p(\Phi_k(\boldsymbol{e})|x,\pi_0)}\left[\mathbb{E}_{p(\boldsymbol{r}|x,\Phi_k(\boldsymbol{e}))}\left[\boldsymbol{r}(k)^2\right]\mathbb{V}_{\pi_0(\Phi_k(\boldsymbol{a})|x,\Phi_k(\boldsymbol{e}))}\left[w_{\Phi_k}(x,\boldsymbol{a})\right]\right]$$

770     $\geq 0$

772 where we change the notation for the total sum in the ranking embedding spaces and ranking action spaces in Eq.(7) and (8).

$$\sum_{\boldsymbol{e} \in \Pi(\mathcal{E})} p(\boldsymbol{e}|x, \boldsymbol{a}) = \sum_{e_1 \in \mathcal{E}} \cdots \sum_{e_K \in \mathcal{E}} p(e_1, \cdots, e_K|x, \boldsymbol{a}) = \sum_{\Phi_k(\boldsymbol{e})} \sum_{\Phi_k^c(\boldsymbol{e})} p(\Phi_k(\boldsymbol{e}), \Phi_k^c(\boldsymbol{e})|x, \boldsymbol{a})$$

$$\sum_{\boldsymbol{a} \in \Pi(\mathcal{A})} \pi_0(\boldsymbol{a}|x) = \sum_{a_1 \in \mathcal{A}_1} \cdots \sum_{a_K \in \mathcal{A}_K} \pi_0(a_1, \cdots, a_K|x) = \sum_{\Phi_k(\boldsymbol{a})} \sum_{\Phi_k^c(\boldsymbol{a})} \underbrace{\pi_0(\Phi_k(\boldsymbol{a}), \Phi_k^c(\boldsymbol{a})|x)}_{=\pi_0(\Phi_k(\boldsymbol{a})|x)\pi_0(\Phi_k^c(\boldsymbol{a})|x, \Phi_k(\boldsymbol{a}))}$$

782 ① in Eq.(9) can be rewritten as the variance with respect to $w_{\Phi_k}(x, \boldsymbol{a})$ as follows:

$$① = \sum_{\Phi_k(\boldsymbol{a})} \pi_0(\Phi_k(\boldsymbol{a})|x, \Phi_k(\boldsymbol{e}))w_{\Phi_k}^2(x, \boldsymbol{a}) - w_{\Phi_k}^2(x, \boldsymbol{e}) \underbrace{\sum_{\Phi_k(\boldsymbol{a})} \pi_0(\Phi_k(\boldsymbol{a})|x, \Phi_k(\boldsymbol{e}))}_{=1}$$

$$= \sum_{\Phi_k(\boldsymbol{a})} \pi_0(\Phi_k(\boldsymbol{a})|x, \Phi_k(\boldsymbol{e}))w_{\Phi_k}^2(x, \boldsymbol{a}) - w_{\Phi_k}^2(x, \boldsymbol{e})$$

$$= \sum_{\Phi_k(\boldsymbol{a})} \pi_0(\Phi_k(\boldsymbol{a})|x, \Phi_k(\boldsymbol{e}))w_{\Phi_k}^2(x, \boldsymbol{a}) - \left( \sum_{\Phi_k(\boldsymbol{a})} \pi_0(\Phi_k(\boldsymbol{a})|x, \Phi_k(\boldsymbol{e}))w_{\Phi_k}(x, \boldsymbol{a}) \right)^2$$

$$= \mathbb{E}_{\pi_0(\Phi_k(\boldsymbol{a})|x, \Phi_k(\boldsymbol{e}))}\left[ w_{\Phi_k}^2(x, \boldsymbol{a}) \right] - \mathbb{E}_{\pi_0(\Phi_k(\boldsymbol{a})|x, \Phi_k(\boldsymbol{e}))}\left[ w_{\Phi_k}(x, \boldsymbol{a}) \right]^2 = \mathbb{V}_{\pi_0(\Phi_k(\boldsymbol{a})|x, \Phi_k(\boldsymbol{e}))}\left[ w_{\Phi_k}(x, \boldsymbol{a}) \right]$$

$$\square$$

## C.3. Variance Reduction of the GMIPS, except the MSIPS, Compared to the MSIPS

**Theorem C.1.** *(Variance Reduction 2)The GMIPS, except the MSIPS, have the following reduction in variance compared to the MSIPS under Assumptions 3.1 and 3.3. See Appendix C.3 for the proof.*

$$n \left( \mathbb{V}_{p(\mathcal{D})}\left[ \hat{V}_{MSIPS}^{(k)}(\pi; \mathcal{D}) \right] - \mathbb{V}_{p(\mathcal{D})}\left[ \hat{V}_{GMIPS}^{(k)}(\pi; \mathcal{D}) \right] \right) = \mathbb{E}_{x, \Phi_k(\boldsymbol{e}) \sim \pi_0}\left[ w_{\Phi_k}^2(x, \boldsymbol{e})\mathbb{V}_{\Phi_k^c(\boldsymbol{e})}\left[ w_{\Phi_k^c}(x, \boldsymbol{e}) \right]\mathbb{E}_{\boldsymbol{r}}\left[ r(k)^2 \right] \right),$$

where $\Phi_k^c(\boldsymbol{e})$ denotes the complement of $\Phi_k(\boldsymbol{e})$. For instance, if $\Phi_k(\boldsymbol{e}) = \boldsymbol{e}(1{:}k)$, then $\Phi_k^c(\boldsymbol{e})$ becomes $\boldsymbol{e}(k{+}1{:}K)$. $w_{\Phi_k^c}(x, \boldsymbol{e}) = \frac{p(\Phi_k^c(\boldsymbol{e})|x, \pi, \Phi_k(\boldsymbol{e}))}{p(\Phi_k^c(\boldsymbol{e})|x, \pi_0, \Phi_k(\boldsymbol{e}))}$ is a doubly marginalized weight of complement set $\Phi_k^c$. Under Assumptions 3.1 and 3.3, the variance of the GMIPS, apart from MSIPS, is theoretically always smaller than that of the MSIPS. Specifically, the larger the space in $\Phi_k^c(\boldsymbol{e})$(the closer $w_{\Phi_k}(x, \boldsymbol{e})$ is to an independent weight), the greater the reduction in variance in the GMIPS, as it includes the variance of $w_{\Phi_k^c}(x, \boldsymbol{e})$. Furthermore, the longer the rank, the greater is the decrease in variance, as Theorem C.1 applies to all positions. Additionally, other factors such as a larger policy deviation $w_{\Phi_k}^2(x, \boldsymbol{e})$ of $\Phi_k(\boldsymbol{e})$, and the noise of the reward $r(k)^2$ contribute to a considerable variance reduction.

*Proof.* Under Assumptions 3.1 and 3.3, the GMIPS variants are unbiased. The difference in their variance is then attributed to their second moment, which is calculated as follows.

$$n \left( \mathbb{V}_{\mathcal{D}}\left[ \hat{V}_{\text{MSIPS}}^{(k)}(\pi; \mathcal{D}) \right] - \mathbb{V}_{\mathcal{D}}\left[ \hat{V}_{\text{GMIPS}}^{(k)}(\pi; \mathcal{D}) \right] \right)$$

$$= n \left( \mathbb{V}_{\mathcal{D}}\left[ w(x, \boldsymbol{e})r(k) \right] - \mathbb{V}_{\mathcal{D}}\left[ w_{\Phi_k}(x, \boldsymbol{e})r(k) \right] \right)$$

$$= \mathbb{E}_{p(x)\pi_0(\boldsymbol{a}|x)p(\boldsymbol{e}|x, \boldsymbol{a})p(\boldsymbol{r}|x, \boldsymbol{a}, \boldsymbol{e})}\left[ \left( w(x, \boldsymbol{e})r(k) \right)^2 \right] - \mathbb{E}_{p(x)\pi_0(\boldsymbol{a}|x)p(\boldsymbol{e}|x, \boldsymbol{a})p(\boldsymbol{r}|x, \boldsymbol{a}, \boldsymbol{e})}\left[ \left( w_{\Phi_k}(x, \boldsymbol{e})r(k) \right)^2 \right]$$

$$= \mathbb{E}_{p(x)p(e|x,\pi_0)}\left[\left(w^2(x,e) - w^2_{\Phi_k}(x,e)\right)\mathbb{E}_{p(r|x,\Phi_k(e))}\left[r(k)^2\right]\right] \quad \because \text{Assumption 3.3}$$

$$= \mathbb{E}_{p(x)p(e|x,\pi_0)}\left[w^2_{\Phi_k}(x,e)\left(w^2_{\Phi_k^c}(x,e) - 1\right)\mathbb{E}_{p(r|x,e)}\left[r(k)^2\right]\right] \quad \because w(x,e) = w_{\Phi_k}(x,e)w_{\Phi_k^c}(x,e)$$

$$= \mathbb{E}_{p(x)p(\Phi_k(e)|x,\pi_0)}\left[w^2_{\Phi_k}(x,e) \underbrace{\mathbb{E}_{p(\Phi_k^c(e)|x,\pi_0,\Phi_k(e))}\left[w^2_{\Phi_k^c}(x,e) - 1\right]}_{\textcircled{1}} \mathbb{E}_{p(r|x,\Phi_k(e))}\left[r(k)^2\right]\right] \tag{10}$$

$$= \mathbb{E}_{p(x)p(\Phi_k(e)|x,\pi_0)}\left[w^2_{\Phi_k}(x,e)\mathbb{V}_{p(\Phi_k^c(e)|x,\pi_0,\Phi_k(e))}\left[w_{\Phi_k^c}(x,e)\right]\mathbb{E}_{p(r|x,\Phi_k(e))}\left[r(k)^2\right]\right]$$

$$\geq 0$$

where $\textcircled{1}$ in Eq.(10) can be rewritten as the variance with respect to $w_{\Phi_k^c}(x,e)$ as follows:

$$\textcircled{1} = \mathbb{E}_{p(\Phi_k^c(e)|x,\pi_0,\Phi_k(e))}\left[w^2_{\Phi_k^c}(x,e)\right] - \underbrace{\mathbb{E}_{p(\Phi_k^c(e)|x,\pi_0,\Phi_k(e))}\left[w_{\Phi_k^c}(x,e)\right]^2}_{=1} = \mathbb{V}_{p(\Phi_k^c(e)|x,\pi_0,\Phi_k(e))}\left[w_{\Phi_k^c}(x,e)\right]$$

$$\square$$

### C.4. Proof of Theorem 3.8

To prove Theorem 3.8, we first present a lemma.

**Lemma C.2.** *For real-valued, bounded functions* $K \in \mathbb{N}, f : \mathbb{N}^K \to \mathbb{R}, g : \mathbb{N}^K \to \mathbb{R}, h : \mathbb{N}^K \to \mathbb{R}, c \in \mathbb{R}$ *where* $\sum_{s\in[m]} g(\boldsymbol{a}_s) = 1$, *we have*

$$\sum_{s\in[m]} f(\boldsymbol{a}_s)g(\boldsymbol{a}_s)\left(h(\boldsymbol{a}_s) - c\sum_{t\in[m]} g(\boldsymbol{a}_t)h(\boldsymbol{a}_t)\right)$$

$$= (1-c)\sum_{s\in[m]} f(\boldsymbol{a}_s)g(\boldsymbol{a}_s)h(\boldsymbol{a}_s) + c\sum_{s<t\leq m} g(\boldsymbol{a}_s)g(\boldsymbol{a}_t)\left(h(\boldsymbol{a}_s) - h(\boldsymbol{a}_t)\right)\left(f(\boldsymbol{a}_s) - f(\boldsymbol{a}_t)\right) \tag{11}$$

*Proof.* We prove this lemma using mathematical induction. First, we show the case for $m = 2$ below.

$$f(\boldsymbol{a}_1)g(\boldsymbol{a}_1)\left(h(\boldsymbol{a}_1) - c\left(g(\boldsymbol{a}_1)h(\boldsymbol{a}_1) + g(\boldsymbol{a}_2)h(\boldsymbol{a}_2)\right)\right) + f(\boldsymbol{a}_2)g(\boldsymbol{a}_2)\left(h(\boldsymbol{a}_2) - c\left(g(\boldsymbol{a}_1)h(\boldsymbol{a}_1) + g(\boldsymbol{a}_2)h(\boldsymbol{a}_2)\right)\right)$$

$$= f(\boldsymbol{a}_1)g(\boldsymbol{a}_1)h(\boldsymbol{a}_1) - cf(\boldsymbol{a}_1)g(\boldsymbol{a}_1)\left(g(\boldsymbol{a}_1)h(\boldsymbol{a}_1) + g(\boldsymbol{a}_2)h(\boldsymbol{a}_2)\right)$$
$$\quad + f(\boldsymbol{a}_2)g(\boldsymbol{a}_2)h(\boldsymbol{a}_2) - cf(\boldsymbol{a}_2)g(\boldsymbol{a}_2)\left(g(\boldsymbol{a}_1)h(\boldsymbol{a}_1) + g(\boldsymbol{a}_2)h(\boldsymbol{a}_2)\right)$$

$$= f(\boldsymbol{a}_1)g(\boldsymbol{a}_1)h(\boldsymbol{a}_1) - cf(\boldsymbol{a}_1)g(\boldsymbol{a}_1)\left((1 - g(\boldsymbol{a}_2))h(\boldsymbol{a}_1) + g(\boldsymbol{a}_2)h(\boldsymbol{a}_2)\right)$$
$$\quad + f(\boldsymbol{a}_2)g(\boldsymbol{a}_2)h(\boldsymbol{a}_2) - cf(\boldsymbol{a}_2)g(\boldsymbol{a}_2)\left(g(\boldsymbol{a}_1)h(\boldsymbol{a}_1) + (1 - g(\boldsymbol{a}_1))h(\boldsymbol{a}_2)\right)$$

$$= (1-c)f(\boldsymbol{a}_1)g(\boldsymbol{a}_1)h(\boldsymbol{a}_1) + cf(\boldsymbol{a}_1)g(\boldsymbol{a}_1)g(\boldsymbol{a}_2)(h(\boldsymbol{a}_1) - h(\boldsymbol{a}_2))$$
$$\quad + (1-c)f(\boldsymbol{a}_2)g(\boldsymbol{a}_2)h(\boldsymbol{a}_2) - cf(\boldsymbol{a}_2)g(\boldsymbol{a}_1)g(\boldsymbol{a}_2)(h(\boldsymbol{a}_1) - h(\boldsymbol{a}_2))$$

$$= (1-c)(f(\boldsymbol{a}_1)g(\boldsymbol{a}_1)h(\boldsymbol{a}_1) + f(\boldsymbol{a}_2)g(\boldsymbol{a}_2)h(\boldsymbol{a}_2)) + cg(\boldsymbol{a}_1)g(\boldsymbol{a}_2)(h(\boldsymbol{a}_1) - h(\boldsymbol{a}_2))(f(\boldsymbol{a}_1) - f(\boldsymbol{a}_2))$$

Note that $g(\boldsymbol{a}_1) + g(\boldsymbol{a}_2) = 1$ from the statement. Next, we assume Eq.(11) is true for the case when $m = k - 1$ and show that it is also true for the case when $m = k$. First, note that

$$(1-c) \sum_{s \in [k]} f(\boldsymbol{a}_s) g(\boldsymbol{a}_s) h(\boldsymbol{a}_s) + c \sum_{s < t \leq k} g(\boldsymbol{a}_s) g(\boldsymbol{a}_t) \left( h(\boldsymbol{a}_s) - h(\boldsymbol{a}_t) \right) \left( f(\boldsymbol{a}_s) - f(\boldsymbol{a}_t) \right)$$

$$= (1-c) \sum_{s \in [k-1]} f(\boldsymbol{a}_s) g(\boldsymbol{a}_s) h(\boldsymbol{a}_s) + c \sum_{s < t \leq k-1} g(\boldsymbol{a}_s) g(\boldsymbol{a}_t) \left( h(\boldsymbol{a}_s) - h(\boldsymbol{a}_t) \right) \left( f(\boldsymbol{a}_s) - f(\boldsymbol{a}_t) \right)$$

$$+ (1-c) f(\boldsymbol{a}_k) g(\boldsymbol{a}_k) h(\boldsymbol{a}_k) + c \sum_{s \in [k-1]} g(\boldsymbol{a}_s) g(\boldsymbol{a}_k) \left( h(\boldsymbol{a}_s) - h(\boldsymbol{a}_k) \right) \left( f(\boldsymbol{a}_s) - f(\boldsymbol{a}_k) \right)$$

Then, we have

$$\sum_{s \in [k]} f(\boldsymbol{a}_s) g(\boldsymbol{a}_s) \left( h(\boldsymbol{a}_s) - c \sum_{t \in [k]} g(\boldsymbol{a}_t) h(\boldsymbol{a}_t) \right)$$

$$= \sum_{s \in [k-1]} f(\boldsymbol{a}_s) g(\boldsymbol{a}_s) \left( h(\boldsymbol{a}_s) - c \sum_{t \in [k]} g(\boldsymbol{a}_t) h(\boldsymbol{a}_t) \right) + f(\boldsymbol{a}_k) g(\boldsymbol{a}_k) \left( h(\boldsymbol{a}_k) - c \sum_{t \in [k]} g(\boldsymbol{a}_t) h(\boldsymbol{a}_t) \right)$$

$$= \sum_{s \in [k-1]} f(\boldsymbol{a}_s) g(\boldsymbol{a}_s) \left( \left( h(\boldsymbol{a}_s) - c \sum_{t \in [k-1]} g(\boldsymbol{a}_t) h(\boldsymbol{a}_t) \right) - c g(\boldsymbol{a}_k) h(\boldsymbol{a}_k) \right) + f(\boldsymbol{a}_k) g(\boldsymbol{a}_k) h(\boldsymbol{a}_k)$$

$$- c f(\boldsymbol{a}_k) g(\boldsymbol{a}_k) \sum_{s \in [k]} g(\boldsymbol{a}_s) h(\boldsymbol{a}_k)$$

$$= \sum_{s \in [k-1]} f(\boldsymbol{a}_s) g(\boldsymbol{a}_s) \left( h(\boldsymbol{a}_s) - c \sum_{t \in [k-1]} g(\boldsymbol{a}_t) h(\boldsymbol{a}_t) \right) - c g(\boldsymbol{a}_k) h(\boldsymbol{a}_k) \sum_{s \in [k-1]} f(\boldsymbol{a}_s) g(\boldsymbol{a}_s)$$

$$+ f(\boldsymbol{a}_k) g(\boldsymbol{a}_k) h(\boldsymbol{a}_k) - c f(\boldsymbol{a}_k) g(\boldsymbol{a}_k) \sum_{s \in [k]} g(\boldsymbol{a}_s) h(\boldsymbol{a}_s)$$

$$= \sum_{s \in [k-1]} f(\boldsymbol{a}_s) g(\boldsymbol{a}_s) \left( h(\boldsymbol{a}_s) - c \sum_{t \in [k-1]} g(\boldsymbol{a}_t) h(\boldsymbol{a}_t) \right) - c g(\boldsymbol{a}_k) h(\boldsymbol{a}_k) \sum_{s \in [k-1]} f(\boldsymbol{a}_s) g(\boldsymbol{a}_s)$$

$$+ f(\boldsymbol{a}_k) g(\boldsymbol{a}_k) h(\boldsymbol{a}_k) - c f(\boldsymbol{a}_k) g(\boldsymbol{a}_k) \sum_{s \in [k-1]} g(\boldsymbol{a}_s) h(\boldsymbol{a}_s) - c f(\boldsymbol{a}_k) g(\boldsymbol{a}_k) g(\boldsymbol{a}_k) h(\boldsymbol{a}_k)$$

$$= (1 - g(\boldsymbol{a}_k)) \sum_{s \in [k-1]} f(\boldsymbol{a}_s) \tilde{g}(\boldsymbol{a}_s) \left( (1 - g(\boldsymbol{a}_k)) \left( h(\boldsymbol{a}_s) - c \sum_{t \in [k-1]} \tilde{g}(\boldsymbol{a}_t) h(\boldsymbol{a}_t) \right) + g(\boldsymbol{a}_k) h(\boldsymbol{a}_s) \right)$$

$$- c g(\boldsymbol{a}_k) h(\boldsymbol{a}_k) \sum_{s \in [k-1]} f(\boldsymbol{a}_s) g(\boldsymbol{a}_s) + f(\boldsymbol{a}_k) g(\boldsymbol{a}_k) h(\boldsymbol{a}_k) - c f(\boldsymbol{a}_k) g(\boldsymbol{a}_k) \sum_{s \in [k-1]} g(\boldsymbol{a}_s) h(\boldsymbol{a}_s)$$

$$- c f(\boldsymbol{a}_k) g(\boldsymbol{a}_k) h(\boldsymbol{a}_k) \left( 1 - \sum_{s \in [k-1]} g(\boldsymbol{a}_s) \right)$$

$$= (1 - g(\boldsymbol{a}_k))^2 \sum_{s \in [k-1]} f(\boldsymbol{a}_s) \tilde{g}(\boldsymbol{a}_s) \left( h(\boldsymbol{a}_s) - c \sum_{t \in [k-1]} \tilde{g}(\boldsymbol{a}_t) h(\boldsymbol{a}_t) \right) + g(\boldsymbol{a}_k) \sum_{s \in [k-1]} f(\boldsymbol{a}_s) g(\boldsymbol{a}_s) h(\boldsymbol{a}_s)$$

$$- c g(\boldsymbol{a}_k) h(\boldsymbol{a}_k) \sum_{s \in [k-1]} f(\boldsymbol{a}_s) g(\boldsymbol{a}_s) - c f(\boldsymbol{a}_k) g(\boldsymbol{a}_k) \sum_{s \in [k-1]} g(\boldsymbol{a}_s) h(\boldsymbol{a}_s)$$

$$+ (1-c) f(\boldsymbol{a}_k) g(\boldsymbol{a}_k) h(\boldsymbol{a}_k) + c f(\boldsymbol{a}_k) g(\boldsymbol{a}_k) h(\boldsymbol{a}_k) \sum_{s \in [k-1]} g(\boldsymbol{a}_s) \tag{12}$$

where we use $g(\boldsymbol{a}_k) = 1 - \sum_{s\in[k-1]} g(\boldsymbol{a}_s)$ and define $\tilde{g}(\boldsymbol{a}_s) := g(\boldsymbol{a}_s)/(\sum_{s\in[k-1]} g(\boldsymbol{a}_s)) = g(\boldsymbol{a}_s)/(1 - g(\boldsymbol{a}_k))$.

The first term of Eq.(12) includes the case when $m = k-1$; thus, we have the following from the assumption of induction.

$$
(1 - g(\boldsymbol{a}_k))^2 \sum_{s\in[k-1]} f(\boldsymbol{a}_s)\tilde{g}(\boldsymbol{a}_s) \left( h(\boldsymbol{a}_s) - c \sum_{t\in[k-1]} \tilde{g}(\boldsymbol{a}_t)h(\boldsymbol{a}_t) \right)
$$

$$
= (1 - g(\boldsymbol{a}_k))^2 \left( (1-c) \sum_{s\in[k-1]} f(\boldsymbol{a}_s)\tilde{g}(\boldsymbol{a}_s)h(\boldsymbol{a}_s) + c \sum_{s<t\leq k-1} \tilde{g}(\boldsymbol{a}_s)\tilde{g}(\boldsymbol{a}_t)\left(h(\boldsymbol{a}_s) - h(\boldsymbol{a}_t)\right)\left(f(\boldsymbol{a}_s) - f(\boldsymbol{a}_t)\right) \right)
$$

$$
= (1 - g(\boldsymbol{a}_k))(1-c) \sum_{s\in[k-1]} f(\boldsymbol{a}_s)g(\boldsymbol{a}_s)h(\boldsymbol{a}_s) + c \sum_{s<t\leq k-1} g(\boldsymbol{a}_s)g(\boldsymbol{a}_t)\left(h(\boldsymbol{a}_s) - h(\boldsymbol{a}_t)\right)\left(f(\boldsymbol{a}_s) - f(\boldsymbol{a}_t)\right)
$$

Note that $\sum_{s\in[k-1]} \tilde{g}(\boldsymbol{a}_s) = 1$. Rearranging the remaining terms of Eq.(12) yields

$$
\sum_{s\in[k]} f(\boldsymbol{a}_s)g(\boldsymbol{a}_s) \left( h(\boldsymbol{a}_s) - c \sum_{t\in[k]} g(\boldsymbol{a}_t)h(\boldsymbol{a}_t) \right)
$$

$$
= (1-c) \sum_{s\in[k-1]} f(\boldsymbol{a}_s)g(\boldsymbol{a}_s)h(\boldsymbol{a}_s) + c \sum_{s<t\leq k-1} g(\boldsymbol{a}_s)g(\boldsymbol{a}_t)\left(h(\boldsymbol{a}_s) - h(\boldsymbol{a}_t)\right)\left(f(\boldsymbol{a}_s) - f(\boldsymbol{a}_t)\right)
$$

$$
+ (1-c)f(\boldsymbol{a}_k)g(\boldsymbol{a}_k)h(\boldsymbol{a}_k) + c \sum_{s\in[k-1]} g(\boldsymbol{a}_s)g(\boldsymbol{a}_k)\left(h(\boldsymbol{a}_s) - h(\boldsymbol{a}_k)\right)\left(f(\boldsymbol{a}_s) - f(\boldsymbol{a}_k)\right)
$$

Implying that the case when $m = k$ is true if the case when $m = k-1$ is true.

$\square$

We then use the above Lemma to prove Theorem 3.8.

*Proof.*

$\mathrm{Bias}\left( \hat{V}_{\mathrm{GMIPS}}^{(k)}(\pi; \mathcal{D}) \right)$

$$
= \mathbb{E}_{p(x)\pi_0(\boldsymbol{a}|x)p(\boldsymbol{e}|x,\boldsymbol{a})p(\boldsymbol{r}|x,\boldsymbol{a},\boldsymbol{e})}\left[ w_{\Phi_k}(x, \boldsymbol{e})\boldsymbol{r}(k) \right] - V^{(k)}(\pi)
$$

$$
= \mathbb{E}_{p(x)\pi_0(\boldsymbol{a}|x)p(\boldsymbol{e}|x,\boldsymbol{a})}\left[ w_{\Phi_k}(x, \boldsymbol{e})q_k(x, \boldsymbol{a}, \boldsymbol{e}) \right] - \mathbb{E}_{p(x)\pi(\boldsymbol{a}|x)p(\boldsymbol{e}|x,\boldsymbol{a})}\left[ q_k(x, \boldsymbol{a}, \boldsymbol{e}) \right]
$$

$$
= \mathbb{E}_{p(x)}\left[ \sum_{\boldsymbol{a}\in\Pi(\mathcal{A})} \pi_0(\boldsymbol{a}|x) \sum_{\boldsymbol{e}\in\Pi(\mathcal{E})} p(\boldsymbol{e}|x,\boldsymbol{a})w_{\Phi_k}(x, \boldsymbol{e})q_k(x, \boldsymbol{a}, \boldsymbol{e}) \right] - \mathbb{E}_{p(x)}\left[ \sum_{\boldsymbol{a}\in\Pi(\mathcal{A})} \pi(\boldsymbol{a}|x) \sum_{\boldsymbol{e}\in\Pi(\mathcal{E})} p(\boldsymbol{e}|x,\boldsymbol{a})q_k(x, \boldsymbol{a}, \boldsymbol{e}) \right]
$$

$$
= \mathbb{E}_{p(x)}\left[ \sum_{\boldsymbol{a}\in\Pi(\mathcal{A})} \pi_0(\boldsymbol{a}|x) \sum_{\boldsymbol{e}\in\Pi(\mathcal{E})} \frac{\pi_0(\boldsymbol{a}|x,\boldsymbol{e})p(\boldsymbol{e}|x,\pi_0)}{\pi_0(\boldsymbol{a}|x)}w_{\Phi_k}(x, \boldsymbol{e})q_k(x, \boldsymbol{a}, \boldsymbol{e}) \right]
$$

$$
- \mathbb{E}_{p(x)}\left[ \sum_{\boldsymbol{a}\in\Pi(\mathcal{A})} \pi(\boldsymbol{a}|x) \sum_{\boldsymbol{e}\in\Pi(\mathcal{E})} \frac{\pi_0(\boldsymbol{a}|x,\boldsymbol{e})p(\boldsymbol{e}|x,\pi_0)}{\pi_0(\boldsymbol{a}|x)}q_k(x, \boldsymbol{a}, \boldsymbol{e}) \right] \quad \because p(\boldsymbol{e}|x,\boldsymbol{a}) = \frac{\pi_0(\boldsymbol{a}|x,\boldsymbol{e})p(\boldsymbol{e}|x,\pi_0)}{\pi_0(\boldsymbol{a}|x)}
$$

$$
= \mathbb{E}_{p(x)}\left[ \sum_{\boldsymbol{e}\in\Pi(\mathcal{E})} p(\boldsymbol{e}|x,\pi_0)w_{\Phi_k}(x, \boldsymbol{e}) \sum_{\boldsymbol{a}\in\Pi(\mathcal{A})} \pi_0(\boldsymbol{a}|x,\boldsymbol{e})q_k(x, \boldsymbol{a}, \boldsymbol{e}) \right]
$$

$$- \mathbb{E}_{p(x)} \left[ \sum_{e \in \Pi(\mathcal{E})} p(e|x, \pi_0) \sum_{a \in \Pi(\mathcal{A})} w(x, a) \pi_0(a|x, e) q_k(x, a, e) \right]$$

$$= \mathbb{E}_{p(x)p(e|x,\pi_0)} \left[ w_{\Phi_k}(x, e) \sum_{a \in \Pi(\mathcal{A})} \pi_0(a|x, e) q_k(x, a, e) \right] - \mathbb{E}_{p(x)p(e|x,\pi_0)} \left[ \sum_{a \in \Pi(\mathcal{A})} w(x, a) \pi_0(a|x, e) q_k(x, a, e) \right]$$

$$= \mathbb{E}_{p(x)p(e|x,\pi_0)} \left[ w(x, e) w_{\Phi_k^c}^{-1}(x, e) \sum_{a \in \Pi(\mathcal{A})} \pi_0(a|x, e) q_k(x, a, e) \right]$$

$$- \mathbb{E}_{p(x)p(e|x,\pi_0)} \left[ \sum_{a \in \Pi(\mathcal{A})} w(x, a) \pi_0(a|x, e) q_k(x, a, e) \right] \quad \because w_{\Phi_k}(x, e) = w(x, e) w_{\Phi_k^c}^{-1}(x, e)$$

$$= \mathbb{E}_{p(x)p(e|x,\pi_0)} \left[ \sum_{a \in \Pi(\mathcal{A})} w(x, a) \pi_0(a|x, e) w_{\Phi_k^c}^{-1}(x, e) \sum_{b \in \Pi(\mathcal{A})} \pi_0(b|x, e) q_k(x, b, e) \right]$$

$$- \mathbb{E}_{p(x)p(e|x,\pi_0)} \left[ \sum_{a \in \Pi(\mathcal{A})} w(x, a) \pi_0(a|x, e) q_k(x, a, e) \right] \quad \because w(x, e) = \sum_{a \in \Pi(\mathcal{A})} w(x, a) \pi_0(a|x, e)$$

$$= \mathbb{E}_{p(x)p(e|x,\pi_0)} \left[ \sum_{a \in \Pi(\mathcal{A})} w(x, a) \pi_0(a|x, e) \left( w_{\Phi_k^c}^{-1}(x, e) \left( \sum_{b \in \Pi(\mathcal{A})} \pi_0(b|x, e) q_k(x, b, e) \right) - q_k(x, a, e) \right) \right] \quad (13)$$

We apply Lemma C.2 to Eq.(13). Setting $f(a) = w(\cdot, a), \quad g(a) = \pi_0(a|\cdot, \cdot), \quad h(a) = q_k(\cdot, a, \cdot), \quad c = w_{\Phi_k^c}^{-1}(\cdot, \cdot)$. We then have the following bias:

$$\mathrm{Bias} \left( \hat{V}_{\mathrm{GMIPS}}^{(k)}(\pi; \mathcal{D}) \right)$$

$$= \mathbb{E}_{p(x)p(e|x,\pi_0)} \left[ \left( w_{\Phi_k^c}^{-1}(x, e) - 1 \right) \sum_{a \in \Pi(\mathcal{A})} w(x, a) \pi_0(a|x, e) q_k(x, a, e) \right]$$

$$+ \mathbb{E}_{p(x)p(e|x,\pi_0)} \left[ w_{\Phi_k^c}^{-1}(x, e) \sum_{s < t \leq |\Pi(\mathcal{A})|} \pi_0(a_s|x, e) \pi_0(a_t|x, e) \left( q_k(x, a_s, e) - q_k(x, a_t, e) \right) \left( w(x, a_t) - w(x, a_s) \right) \right]$$

$$= \mathbb{E}_{p(x)} \left[ \sum_{a \in \Pi(\mathcal{A})} \pi(a|x) \sum_{e \in \Pi(\mathcal{E})} \frac{\pi_0(a|x, e) p(e|x, \pi_0)}{\pi_0(a|x)} \left( w_{\Phi_k^c}^{-1}(x, e) - 1 \right) q_k(x, a, e) \right]$$

$$+ \mathbb{E}_{p(x)p(e|x,\pi_0)} \left[ w_{\Phi_k^c}^{-1}(x, e) \sum_{s < t \leq |\Pi(\mathcal{A})|} \pi_0(a_s|x, e) \pi_0(a_t|x, e) \left( q_k(x, a_s, e) - q_k(x, a_t, e) \right) \left( w(x, a_t) - w(x, a_s) \right) \right]$$

$$= \mathbb{E}_{p(x)\pi(a|x)p(e|x,a)} \left[ \left( w_{\Phi_k^c}^{-1}(x, e) - 1 \right) q_k(x, a, e) \right]$$

$$+ \mathbb{E}_{p(x)p(e|x,\pi_0)} \left[ w_{\Phi_k^c}^{-1}(x, e) \sum_{s < t \leq |\Pi(\mathcal{A})|} \pi_0(a_s|x, e) \pi_0(a_t|x, e) \left( q_k(x, a_s, e) - q_k(x, a_t, e) \right) \left( w(x, a_t) - w(x, a_s) \right) \right]$$

$$\because p(e|x, a) = \frac{\pi_0(a|x, e) p(e|x, \pi_0)}{\pi_0(a|x)}$$

$$\square$$

**C.5. Bias of the GMIPS when Assumption 3.3. does not Hold under Assumptions 3.1. and 3.2.**

**Theorem C.3.** *(Bias of GMIPS 2 )Under Assumption 3.1, the GMIPS, except the MSIPS, have the following bias when Assumption 3.2 holds but Assumption 3.3 does not hold.*

$$Bias\left(\hat{V}_{GMIPS}^{(k)}(\pi;\mathcal{D})\right) = \mathbb{E}_{x,\boldsymbol{e}\sim\pi}\left[\left(w_{\Phi_k^c}^{-1}(x,\boldsymbol{e}) - 1\right)q_k(x,\boldsymbol{e})\right],$$

where $\Delta w_{\Phi_k^c}^{-1}(x,\boldsymbol{e}) := w_{\Phi_k^c}^{-1}(x,\boldsymbol{e}) - 1$ is a degree of divergence between logging and target policy in the complement set $\Phi_k^c(\boldsymbol{e})$ for each position. The larger the space in $\Phi_k^c(\boldsymbol{e})$, the greater the bias becomes if assumption 3.3 does not hold. Specifically, this suggests that the bias increases as $w_{\Phi_k}(x,\boldsymbol{e})$ uses approaches with independent weights.

*Proof.*

$$\mathbb{E}_{\mathcal{D}}\left[\hat{V}_{\text{GMIPS}}^{(k)}(\pi;\mathcal{D})\right] = \mathbb{E}_{p(x)\pi_0(\boldsymbol{a}|x)p(\boldsymbol{e}|x,\boldsymbol{a})p(\boldsymbol{r}|x,\boldsymbol{a},\boldsymbol{e})}\left[w_{\Phi_k}(x,\boldsymbol{e})\boldsymbol{r}(k)\right]$$

$$= \mathbb{E}_{p(x)}\left[\sum_{\boldsymbol{a}\in\Pi(\mathcal{A})}\pi_0(\boldsymbol{a}|x)\sum_{\boldsymbol{e}\in\Pi(\mathcal{E})}p(\boldsymbol{e}|x,\boldsymbol{a})w_{\Phi_k}(x,\boldsymbol{e})q_k(x,\boldsymbol{e})\right] \quad \because \text{Assumption 3.2}$$

$$= \mathbb{E}_{p(x)}\left[\sum_{\boldsymbol{e}\in\Pi(\mathcal{E})}p(\boldsymbol{e}|x,\pi_0)w_{\Phi_k}(x,\boldsymbol{e})q_k(x,\boldsymbol{e})\right] \quad \because p(\boldsymbol{e}|x,\pi_0) = \sum_{\boldsymbol{a}\in\Pi(\mathcal{A})}\pi_0(\boldsymbol{a}|x)p(\boldsymbol{e}|x,\boldsymbol{a})$$

$$= \mathbb{E}_{p(x)}\left[\sum_{\boldsymbol{e}\in\Pi(\mathcal{E})}p(\boldsymbol{e}|x,\pi)\frac{p(\boldsymbol{e}|x,\pi_0)}{p(\boldsymbol{e}|x,\pi)}\frac{p(\Phi_k(\boldsymbol{e})|x,\pi)}{p(\Phi_k(\boldsymbol{e})|x,\pi_0)}q_k(x,\boldsymbol{e})\right]$$

$$= \mathbb{E}_{p(x)p(\boldsymbol{e}|x,\pi)}\left[\frac{p(\Phi_k^c(\boldsymbol{e})|x,\pi_0,\Phi_k(\boldsymbol{e}))}{p(\Phi_k^c(\boldsymbol{e})|x,\pi,\Phi_k(\boldsymbol{e}))}q_k(x,\boldsymbol{e})\right] \quad \because p(\Phi_k^c(\boldsymbol{e})|x,\pi,\Phi_k(\boldsymbol{e})) = \frac{p(\boldsymbol{e}|x,\pi)}{p(\Phi_k(\boldsymbol{e})|x,\pi)}$$

$$= \mathbb{E}_{p(x)p(\boldsymbol{e}|x,\pi)}\left[w_{\Phi_k^c}^{-1}(x,\boldsymbol{e})q_k(x,\boldsymbol{e})\right]$$

Therefore, we have the following bias:

$$\text{Bias}\left(\hat{V}_{\text{GMIPS}}^{(k)}(\pi;\mathcal{D})\right) = \mathbb{E}_{p(x)p(\boldsymbol{e}|x,\pi)}\left[w_{\Phi_k^c}^{-1}(x,\boldsymbol{e})q_k(x,\boldsymbol{e})\right] - V^{(k)}(\pi)$$

$$= \mathbb{E}_{p(x)p(\boldsymbol{e}|x,\pi)}\left[w_{\Phi_k^c}^{-1}(x,\boldsymbol{e})q_k(x,\boldsymbol{e})\right] - \mathbb{E}_{p(x)\pi(\boldsymbol{a}|x)p(\boldsymbol{e}|x,\boldsymbol{a})}[q_k(x,\boldsymbol{a},\boldsymbol{e})]$$

$$= \mathbb{E}_{x,\boldsymbol{e}\sim\pi}\left[\left(w_{\Phi_k^c}^{-1}(x,\boldsymbol{e}) - 1\right)q_k(x,\boldsymbol{e})\right] \quad \because \text{Assumption 3.2}, p(\boldsymbol{e}|x,\pi) = \sum_{\boldsymbol{a}\in\Pi(\mathcal{A})}\pi(\boldsymbol{a}|x)p(\boldsymbol{e}|x,\boldsymbol{a})$$

$\square$

C.5.1. ADDITIONAL ANALYSIS

**Proposition C.4.** *If Assumption 3.2 holds, the bias in Theorem 3.8 is equivalent to that in Theorem C.3.*

*Proof.*

$$\text{Bias}\left(\hat{V}_{\text{GMIPS}}^{(k)}(\pi;\mathcal{D})\right)$$

$$= \mathbb{E}_{p(x)\pi(\boldsymbol{a}|x)p(\boldsymbol{e}|x,\boldsymbol{a})}\left[\left(w_{\Phi_k^c}^{-1}(x,\boldsymbol{e}) - 1\right)q_k(x,\boldsymbol{a},\boldsymbol{e})\right]$$

$$+ \mathbb{E}_{p(x)p(\boldsymbol{e}|x,\pi_0)}\left[w_{\Phi_k^c}^{-1}(x,\boldsymbol{e})\sum_{s<t\le|\Pi(\mathcal{A})|}\pi_0(\boldsymbol{a}_s|x,\boldsymbol{e})\pi_0(\boldsymbol{a}_t|x,\boldsymbol{e})\Big(q_k(x,\boldsymbol{a}_s,\boldsymbol{e}) - q_k(x,\boldsymbol{a}_t,\boldsymbol{e})\Big)\Big(w(x,\boldsymbol{a}_t) - w(x,\boldsymbol{a}_s)\Big)\right]$$

$$= \mathbb{E}_{p(x)\pi(\boldsymbol{a}|x)p(\boldsymbol{e}|x,\boldsymbol{a})}\left[\left(w_{\Phi_k^c}^{-1}(x,\boldsymbol{e}) - 1\right)q_k(x,\boldsymbol{e})\right] \quad \because \text{Assumption 3.2(i.e., } q_k(x,\boldsymbol{a},\boldsymbol{e}) = q_k(x,\boldsymbol{e}))$$

$\square$

## D. Details of Experimental Setup

Our experiment operates within a Docker container (Merkel et al., 2014) [4] that can be reproduced on any operating system.

### D.1. Data Generation Process of Synthetic Experiments

We generated synthetic data based on the *Open Bandit Pipeline (OBP)* [5] (Saito et al., 2020), and synthetic experimental settings of previous studies (Saito & Joachims, 2022; Kiyohara et al., 2022; 2023). First, we independently generated five-dimensional contexts $x$ from a standard normal distribution. We then generated a distribution of categorical embeddings $p(\boldsymbol{e}|\boldsymbol{a}) = \prod_{k=1}^{K} p(\boldsymbol{e}(k)|\boldsymbol{a}(k))$, which is independent of context $x$ and position $k$ as follows:

$$p(\boldsymbol{e}(k)|\boldsymbol{a}(k)) = \prod_{d=1}^{D} \frac{\exp(\alpha_{\boldsymbol{a}(k),\boldsymbol{e}(k)_d})}{\sum_{e \in \mathcal{E}_d} \exp(\alpha_{\boldsymbol{a}(k),e})}$$

where $D$ is the number of embedding dimensions. We set $K = 5$, $D = 3$, and $|\mathcal{E}_d| = 2$ (i.e., $|\Pi(\mathcal{E})| = 2^{3\times5}$) as default values. $\alpha$ is an unknown variable sampled from a normal distribution. Next, we generated rewards $\boldsymbol{r}$ that satisfy Assumption 3.3. Base reward function $\bar{q}(x, e)$, given $x$ and $e$, is defined as follows:

$$\bar{q}(x, e) = \sum_{d=1}^{D} \eta_d \cdot \sigma(x^\top M x_{e_d} + \theta_x^\top x + \theta_e^\top x_{e_d})$$

where $\eta_d, M, \theta_x, \theta_e$ are unknown parameters that define base reward $\bar{q}$, and $\sigma(x) = 1/(1 + \exp(-x))$ is a sigmoid function. We then incorporate Assumption 3.3 into base reward $\bar{q}$ to generate the expected reward.

$$q_k(x, \boldsymbol{e}) = \bar{q}(x, \boldsymbol{e}(k)) + \begin{cases} \sum_{l \neq k} \mathbb{W}(k, l) & \text{(standard)} \\ \sum_{l < k} \mathbb{W}(k, l) & \text{(cascade)} \\ 0 & \text{(independent)} \end{cases}$$

where $\mathbb{W}(k, l) := |k - l|^{-1}\mathbb{G}(k, l)\bar{q}(x, \boldsymbol{e}(l))$ is an interaction reward function and $\mathbb{G}(\cdot, \cdot) \in [0, 3]$ is a parameter matrix that indicates the extent to which the reward $\boldsymbol{r}(k)$ at position $k$ interacts with other positions $l$. This parameter is generated independently from a uniform distribution. $|k - l|$ is a decay function, indicating that as the distance between the pair of positions $(k, l)$ increases, the interaction value decreases.

Finally, we obtain the reward from normal distribution $\boldsymbol{r}(k) \sim \mathcal{N}(q_k(x, \boldsymbol{e}), \sigma_r^2)$ for each position $k$, where $\sigma_r$ is a reward noise. We set $\sigma_r = 0.5$ as the default value.

Next, we define a logging policy that enables us to express $\pi_0(\boldsymbol{a}|x) = \prod_{k=1}^{K} \pi_0(\boldsymbol{a}(k)|x)$ as follows:

$$\pi_0(\boldsymbol{a}(k)|x) = \frac{\exp(\beta \cdot \bar{q}(x, \boldsymbol{a}(k)))}{\sum_{a \in \mathcal{A}_k} \exp(\beta \cdot \bar{q}(x, a))} \tag{14}$$

where $\bar{q}(x, a) = \mathbb{E}_{p(e|a)}[\bar{q}(x, e)]$ is a base reward function for unique actions and $\beta$ is an inverse temperature parameter. If $\beta > 0$, $\pi_0$ approaches optimality for $\bar{q}(x, a)$, $\beta < 0$, $\pi_0$ approaches anti-optimality for $\bar{q}(x, a)$, and $\beta = 0$, $\pi_0$ becomes a uniform random policy. We set $\beta = -1.0$ and $|\mathcal{A}_k| = 20, \forall k \in [K]$ as default values. We define a unique action set as $\mathcal{A} := \{\mathcal{A}_k\}_{k=1}^{K}$ (i.e. $|\mathcal{A}| = 20 \cdot 5 = 100, |\Pi(\mathcal{A})| = 20^5$). We then obtain $n$ logged data $(x, \boldsymbol{a}, \boldsymbol{e}, \boldsymbol{r}) \sim p(x)\pi_0(\boldsymbol{a}|x)p(\boldsymbol{e}|\boldsymbol{a})p(\boldsymbol{r}|x, \boldsymbol{e})$, and set $n$ to 10,000 as the default value.

Next, we define a target policy that enables us to express $\pi(\boldsymbol{a}|x) = \prod_{k=1}^{K} \pi(\boldsymbol{a}(k)|x)$ as follows:

$$\pi(\boldsymbol{a}(k)|x) = (1 - \epsilon)\,\mathbb{I}\{\boldsymbol{a}(k) = \arg\max_{a \in \mathcal{A}_k} \bar{q}(x, a)\} + \epsilon/|\mathcal{A}_k| \tag{15}$$

---

[4] https://docs.docker.com/desktop/
[5] https://github.com/st-tech/zr-obp

where $\epsilon \in [0,1]$ is an experimental parameter; the closer it is to 0, the more optimal the target policy $\pi$ becomes for $\bar{q}(x,a)$. We set $\epsilon = 0.3$ as the default value.

### D.2. Preprocessing of Real-World Data Experiments

*Table 2.* Statistics for the EUR-Lex4K and RCV1-2K datasets from the Extreme Classification Repository (Bhatia et al., 2016). Note that for the RCV1-2K dataset, the top 1600 training and top 4000 test samples were extracted for the experiment.

|  | train size $n_{\text{train}}$ | test size $n_{\text{test}}$ | dimensions of features | number of labels |
| --- | --- | --- | --- | --- |
| EUR-Lex4K | 15,539 | 3,809 | 5,000 | 3,993 |
| RCV1-2K | 623,847 | 155,962 | 47,236 | 2,456 |

**Setting of rewards that follows diverse user behaviors** We import the experimental setup of (Kiyohara et al., 2023), then redefine following the user behavior distribution given $x$, which is unknown.

$$p(\boldsymbol{c}_z | x) = \frac{\exp(\lambda_z \cdot |\theta_z^\top x|)}{\sum_{z'} \exp(\lambda_{z'} \cdot |\theta_{z'}^\top x|)}$$

where $\boldsymbol{c} \in \{0,1\}^{K \times K}$ is an action–interaction matrix that determines whether compensation for position $k$ is affected by position $l$, $z$ is the name of the user behavior, e.g., chosen from the set {standard, independent, cascade}, $\theta_z$ is a unknown parameter vector per $z$, and generated from the uniform distribution range of $[-1.0, 1.0]$. $\lambda_z$ is a temperature parameter per $z$. We set $\lambda_z = 1.0, \forall z$ as a default value.

Next, we define $\boldsymbol{c}_z$ for the experiments as follows:

- **Standard**: $\boldsymbol{c}_{\text{standard}}(k,l) = 1, \forall k \in [K], l \in [K]$

- **Cascade**: $\boldsymbol{c}_{\text{cascade}}(k,l) = 1, \forall l \leq k$, and $\boldsymbol{c}_{\text{cascade}}(k,l) = 0$, otherwise.

- **Independent**: $\boldsymbol{c}_{\text{independent}}(k,l) = 1, \forall l = k$, and $\boldsymbol{c}_{\text{independent}}(k,l) = 0$, otherwise.

- **Top_2_Cascade**: $\boldsymbol{c}_{\text{top\_2\_cascade}}(k,l) = 1, \forall l \leq 2, l = k$, and $\boldsymbol{c}_{\text{top\_2\_cascade}}(k,l) = 0$, otherwise.

- **Neighbor_1**: $\boldsymbol{c}_{\text{neighbor\_1}}(k,l) = 1, \forall \max(0, k-1) \leq k \leq \min(K, k+1), l = k$, and $\boldsymbol{c}_{\text{neighbor\_1}}(k,l) = 0$, otherwise.

- **Inverse_Cascade**: $\boldsymbol{c}_{\text{inverse\_cascade}}(k,l) = 1, \forall l \geq k$, and $\boldsymbol{c}_{\text{inverse\_cascade}}(k,l) = 0$, otherwise.

- **Random_0**: ( $\boldsymbol{c}_{\text{random\_0}} \sim \text{Be}_{\text{seed=0}}$, or $\boldsymbol{c}_{\text{random\_0}} = 1, \forall l = k$), and $\boldsymbol{c}_{\text{random\_0}}(k,l) = 0$, otherwise.

- **Random_1**: ( $\boldsymbol{c}_{\text{random\_1}} \sim \text{Be}_{\text{seed=1}}$, or $\boldsymbol{c}_{\text{random\_1}} = 1, \forall l = k$), and $\boldsymbol{c}_{\text{random\_1}}(k,l) = 0$, otherwise.

- **Random_2**: ( $\boldsymbol{c}_{\text{random\_2}} \sim \text{Be}_{\text{seed=2}}$, or $\boldsymbol{c}_{\text{random\_2}} = 1, \forall l = k$), and $\boldsymbol{c}_{\text{random\_2}}(k,l) = 0$, otherwise.

We then utilize $\boldsymbol{c} \in \{\boldsymbol{c}_{\text{top\_2\_cascade}}, \boldsymbol{c}_{\text{cascade}}, \boldsymbol{c}_{\text{neighbor\_1}}, \boldsymbol{c}_{\text{inverse\_cascade}}, \boldsymbol{c}_{\text{standard}}, \boldsymbol{c}_{\text{random\_1}}, \boldsymbol{c}_{\text{random\_2}}\}$ to generate rewards, and define the expected reward using base reward function $\bar{q}_k(x, \boldsymbol{a}(k)) = \bar{q}(x,a)$

$$q_k(x, \boldsymbol{a}, \boldsymbol{c}) = \boldsymbol{c}(k,k)\bar{q}_k(x, \boldsymbol{a}(k)) + \sum_{l \neq k} \boldsymbol{c}(k,l)|k-l|^{-1}\mathbb{G}(k,l)\bar{q}(x, \boldsymbol{a}(l))$$

where we set a parameter matrix to $\mathbb{G} \in [0, 15]$ sampled from a uniform distribution. Finally, we can observe binary reward $\boldsymbol{r}(k) \sim Be(\sigma(q_k(x, \boldsymbol{a}, \boldsymbol{c})))$ for each $k$. Regarding the optimization of AIPS(w/UBT), we set the candidate user behavior as $\{\boldsymbol{c}_{\text{independent}}, \boldsymbol{c}_{\text{top\_2\_cascade}}, \boldsymbol{c}_{\text{neighbor\_1}}, \boldsymbol{c}_{\text{cascade}}, \boldsymbol{c}_{\text{random\_0}}\}$ to optimize AIPS(w/UBT). Note that in the synthetic experiments, we set candidate user behavior $\{\boldsymbol{c}_{\text{cascade}}, \boldsymbol{c}_{\text{independent}}, \boldsymbol{c}_{\text{neighbor\_1}}, \boldsymbol{c}_{\text{random\_0}}\}$ to optimize snAIPS(w/UBT).

Regarding the target and logging policies, we apply Eq.(14) to estimate base reward function $\hat{\bar{q}}_k(x, \boldsymbol{a}(k))$, which we use ridge regression for each $k$ as the logging policy. We set $\beta = -1.0$. We then define the target policy by applying Eq.(15) to $\hat{\bar{q}}_k(x, \boldsymbol{a}(k))$. We set $\epsilon = 0.3$.

# E. Embedding Selection of the GMIPS using the SLOPE Algorithm (Su et al., 2020b; Tucker & Lee, 2021)

We conduct the embedding selection for the GMIPS to minimize both bias and variance through a data-driven procedure known as the SLOPE algorithm (Su et al., 2020b; Tucker & Lee, 2021), which is based on Lepski's principle (Lepski & Spokoiny, 1997). By assuming the following monotonicity (Tucker & Lee, 2021), we select an optimally appropriate number of dimensions from the candidate set $\{\theta_m\}_{m=1}^{M}$.

$$(1) \quad \text{Bias}[\hat{V}(\pi; \mathcal{D}, \theta_m)] \leq \text{Bias}[\hat{V}(\pi; \mathcal{D}, \theta_m)], \forall m \in [M]$$

$$(2) \quad \text{CNF}[\hat{V}(\pi; \mathcal{D}, \theta_m)] \geq \text{CNF}[\hat{V}(\pi; \mathcal{D}, \theta_m)], \forall m \in [M]$$

where CNF$[\cdot]$ is a high probability bound on $|\mathbb{E}_{p(\mathcal{D})}[\hat{V}(\pi; \mathcal{D})] - V(\pi)|$, such as concentration inequalities. SLOPE then determines the index of the hyperparameters based on the following criteria:

$$\hat{m} := \max\{j : |\hat{V}(\pi; \mathcal{D}, \theta_j) - \hat{V}(\pi; \mathcal{D}, \theta_m)| \leq \text{CNF}[\hat{V}(\pi; \mathcal{D}, \theta_j)] + (\sqrt{6} - 1)\text{CNF}[\hat{V}(\pi; \mathcal{D}, \theta_m)], \forall m < j\}$$

When the monotonicity assumption holds, SLOPE guarantees that with a probability of at least $1 - \delta$,

$$|\hat{V}(\pi; \mathcal{D}, \theta_{\hat{m}}) - \hat{V}(\pi; \mathcal{D}, \theta_{m^*})| \leq (\sqrt{6} + 3) \leq \min_{m \in [M]} \left(\text{Bias}[\hat{V}(\pi; \mathcal{D}, \theta_m)] + \text{CNF}[\hat{V}(\pi; \mathcal{D}, \theta_m)]\right)$$

For detailed explanations, please refer to (Tucker & Lee, 2021; Saito & Joachims, 2022). We utilized SLOPE to determine the optimal number of dimensions to extract from the first dimension of the embedding, whereas (Saito & Joachims, 2022) optimized the combination of embedding dimensions [6]. Therefore, we achieved a computational efficiency of $\mathcal{O}(D)$, where $D$ is the number of embedding dimensions. From Theorems 3.7 and 3.8, the number of dimensions influences the trade-off between bias and variance. Therefore, it is also effective to apply SLOPE with the number of dimensions as a hyperparameter, rather than relying on a combination of embeddings. Our experiments demonstrate that dimensionality, rather than a combination of embeddings, serves as a sufficiently effective hyperparameter for SLOPE. In this context, we estimate a high probability bound on the deviation CNF$[\cdot]$ based on the Student's t-distribution.

# F. Additional Results of Synthetic Experiments

Here, we briefly discuss experimental results that could not be included in the main text.

**Performance of our estimators with varying noise levels** We varied the noise levels $\sigma_r \in \{0.5, 1.0, 2.0, 4.0, 8.0\}$. The results are presented in Figure 5(a). We observed that our unbiased estimators, as shown in each figure, outperform the existing estimators, specifically on the left and right side of the figure as the reward noise increases. This is because the variance reduction in Theorem 3.7 includes the expected value of $r(k)^2$. This enables the MSIPS and MRIPS to sustain a low MSE even as the reward noise increases.

**Performance of our estimators with varying logging policies** We varied $\beta \in \{-3.0, -1.0, 0.0, 1.0, 3.0\}$ of the logging policy, Eq.(14). That is, as $\beta$ decreases, the deviation from the target policy increases. Conversely, as $\beta$ increases, the deviation from the target policy decreases. The results are presented in Figure 5(b). We observed that our unbiased estimators, as shown in each figure, mostly outperformed the existing estimators as $\beta$ decreases. However, as shown on the left of the figure, the MSE of the MSIPS deteriorates owing to its ranking-wise weights as $\beta$ decreases.

**Performance of our estimators with varying target policies** We varied $\epsilon \in \{0.05, 0.2, 0.4, 0.6, 0.8, 1.0\}$ of the target policy, Eq.(15). That is, as $\epsilon$ decreases, the deviation from the logging policy increases, bringing it closer to the deterministic target policy. Conversely, as $\epsilon$ increases, the target policy approaches the uniform random policy. The results are presented in Figure 5(c). We observed that our unbiased estimators, as shown in each figure, mostly outperformed the existing estimators as $\beta$ decreases. However, the left of the figure shows that the MSE of the MSIPS deteriorates owing to its ranking-wise weights as $\epsilon$ decreases as well as one above experiment.

**Performance of our estimators with varying the number of deficient unique actions** We varied the number of deficient unique actions $\{0, 15, 30, 45, 60\}$. That is, as this number increases, the percentage of cases in which Assumption 2.1

---

[6]Note that our objective is to identify the optimal "ranking embedding," but because the length of the ranking is predetermined, establishing the embedding dimension inherently defines the space for the ranking embedding.

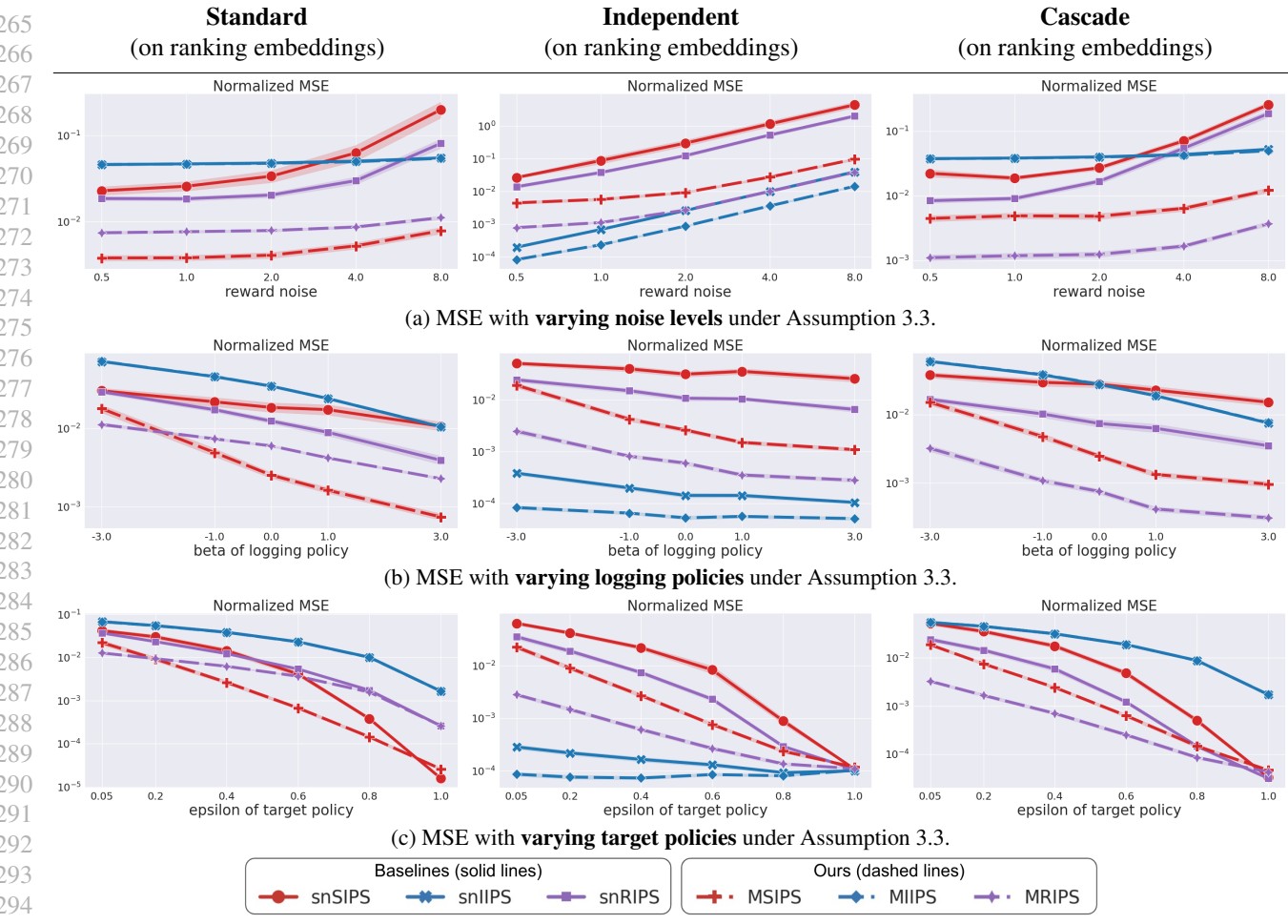

*Figure 5.* Comparison of MSE normalized by $V(\pi)$ under data where the assumption of user behavior model on ranking embeddings are valid. The solid lines are existing estimators and the dashed lines are our proposed estimators. The colors are changed according to the user behavior model assumed on each space. Note that all MSE values are log-scale.

(Common Support) is not satisfied increases. The results are presented in Figure 6. We observed that despite an increasing number of deficient unique actions, the MSE of our estimator remains constant and consistently lower than that of the GIPS, which tends to introduce greater bias. Note that self-normalization (Swaminathan & Joachims, 2015b) was not applied to the GIPS in this experiment. This is because even if Assumption 2.1 is not satisfied, we can still expect our estimator to be unbiased as long as Assumption 3.1 holds.

**Performance of our estimators if Assumption 3.3 does not hold while Assumption 3.2 does hold** We varied the complexity of user behavior. As the complexity increases, the diversity of user behavior in ranking embeddings also expands. Specifically, when the complexity of user behavior is $0.0$, we utilize only independent behavior $c_{\text{independent}}$ on ranking embeddings to generate rewards. Conversely, when the complexity of user behavior reaches $1.0$, we incorporate the following candidate set: $\{c_{\text{independent}}, c_{\text{top\_2\_cascade}}, c_{\text{neighbor\_1}}, c_{\text{cascade}}, c_{\text{inverse\_cascade}}, c_{\text{standard}}\}$. The results are presented in Figure 7. We observed that the MRIPS achieved the lowest MSE in most cases, although the cascade model on ranking embeddings does not hold. Despite snAIPS (w/UBT) adjusting the optimal importance weights according to user context, the MSE deteriorates as user behavior becomes more complex and diverse, which is primarily owing to bias and variance issues associated with large ranking action spaces. Owing to its strong behavior assumptions as the complexity of user behavior increases, the MSE of the MIIPS is greater than that of snAIPS (w/UBT) and snRIPS.

**Performance of our estimators using SLOPE, which selects the optimal embedding dimension** We set $D = 12$ and varied the sample sizes $\{2000, 4000, 8000, 16000, 32000\}$ in the logged data. Here, the MSIPS (w/SLOPE) and MRIPS (w/SLOPE) utilized the importance weights calculated after determining the optimal embedding dimension through SLOPE,

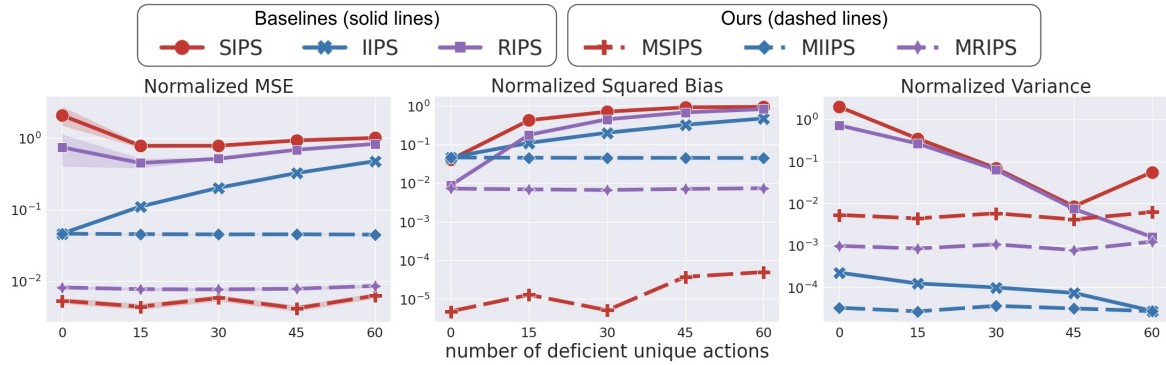

*Figure 6.* MSE, bias, and variance normalized by $V(\pi)$ with **varying number of deficient unique actions** under Assumption 3.1 and 3.2. Note that these are all log-scale.

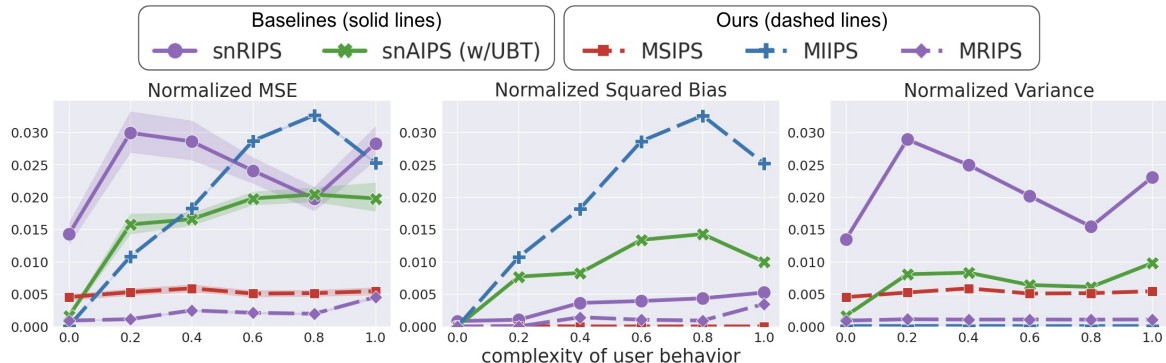

*Figure 7.* MSE, bias, and variance normalized by $V(\pi)$ with **varying complexity of the user behavior** under which Assumption 3.3 does not hold while Assumption 3.1. and 3.2. hold. Note that these are all linear-scale.

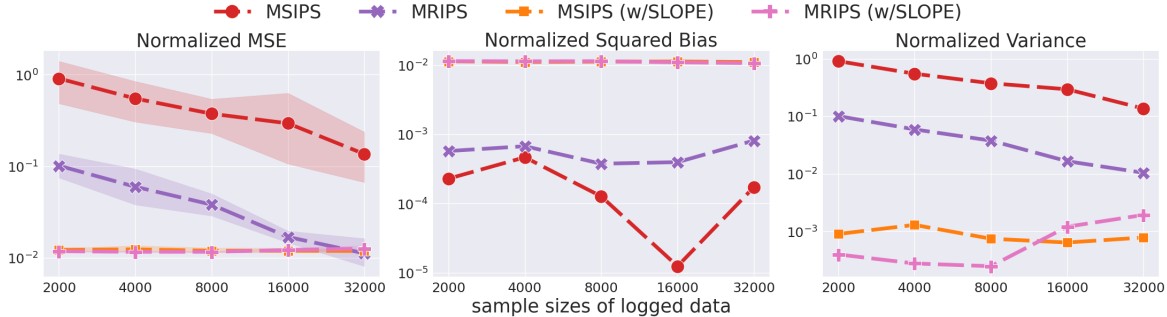

*Figure 8.* MSE, bias, and variance normalized by $V(\pi)$ with **variation in sample sizes of logged data**. Comparison of MSIPS, MRIPS without SLOPE, and MSIPS(w/SLOPE), MRIPS(w/SLOPE). Note that these are all log-scale.

as described in Section E. The results are presented in Figure 8. We observed that the MSEs of the MSIPS(w/SLOPE) and MRIPS(w/SLOPE) were smaller than those of the MSIPS and MRIPS without SLOPE. The latter exhibits a larger MSE owing to the high dimensionality of the embeddings, particularly when the sample size is small. This is because by properly selecting the embedding dimension using only logged data, the MSIPS(w/SLOPE) and MRIPS(w/SLOPE) successfully reduce the variance significantly while permitting some bias. Regarding computational efficiency, this suggests that the MSE can be significantly improved by determining the number of embedding dimensions using SLOPE, rather than optimizing the combination of embeddings as done by (Saito & Joachims, 2022).

