# OpenReview forum: "Off-Policy Evaluation of Ranking Policies for Large Action Spaces via Embeddings and User Behavior Assumption"
_ICML.cc/2025/Conference — Submitted to ICML 2025_

### Official Review · Reviewer_xuP4 · 2025-02-24

**Overall Recommendation:** 3

**Summary:**

This paper addresses off-policy evaluation in ranking contexts when the action space is large, driven by both the number of unique items and the ranking length. Existing estimators often suffer from excessive variance or high bias. The authors propose a Generalized Marginalized IPS (GMIPS) framework that relies on ranking embeddings and assumptions about user behavior in the embedding space (plus a no-direct-effect assumption). They introduce estimators like MSIPS, MIIPS, and MRIPS, offering theoretical guarantees on unbiasedness and variance reduction, along with a bias–variance trade-off analysis when assumptions are partially violated. Experiments on synthetic and real data (EUR-Lex4K and RCV1-2K) show that GMIPS methods can substantially reduce MSE compared to standard IPS-based estimators, and further enhancements come from selecting embedding dimensions via SLOPE.

**Claims And Evidence:**

Key Claims are
- GMIPS estimators (MSIPS, MRIPS, MIIPS) can reduce variance significantly compared to standard IPS-based estimators on the action space (e.g., SIPS, RIPS, or AIPS).


- GMIPS can remain unbiased even when Assumption 2.1 (common support on the action space) is violated, provided the new assumptions (common embedding support, no direct effect, user behavior model on embedding space) hold.


- There is a controllable bias–variance trade-off via double marginalization over embedding subsets: smaller embedding subsets lower variance but risk higher bias if the assumption of no direct effect is violated.

Supporting Evidence
- Theorems in Sections 3 and 4 (especially Theorems 3.7 and 3.8) formally prove the variance reduction and characterize the bias when assumptions are violated.


- A series of synthetic experiments shows consistent MSE reductions for GMIPS-based estimators as sample size, number of unique actions, and ranking length vary, compared to baseline estimators (snSIPS, snRIPS, etc.).

- Real-world data experiments on the EUR-Lex4K and RCV1-2K datasets confirm that GMIPS variants can outperform standard estimators, especially for large action spaces, and further improvement is shown using the SLOPE method to choose embedding dimensions.
All main claims have direct experimental or theoretical backing.

**Essential References Not Discussed:**

The paper cites most of the foundational OPE references (Horvitz & Thompson, Dudík et al., Swaminathan & Joachims, Li et al., Saito & Joachims, etc.) and the relevant ranking-specific works (Kiyohara et al. 2023, McInerney et al. 2020). I do not see a critical missing citation that is central to ranking OPE with large action spaces.

One possible extension could be referencing more closely the slate OPE frameworks (e.g., Swaminathan et al. 2017 or related “pseudo-inverse” approaches), although the paper focuses specifically on the ranking scenario with position-wise rewards. Such references might give broader context but are not strictly required, since the problem setting is quite well-covered by the references they have.

**Experimental Designs Or Analyses:**

In both synthetic and real data, the design (logging policy, target policy, hyperparameter settings) is specified in detail. The experiments are generally valid and consistent with the paper’s claims. One potential consideration is that the real-world transformations (e.g., label embeddings for EUR-Lex4K and RCV1-2K) might not be fully reflective of typical recommendation settings, but the authors do state how these transformations are performed, and such an approach is a common practice in some of the prior literature at least.

**Methods And Evaluation Criteria:**

The authors’ proposed methods by extending marginalization to an embedding space rather than the raw action space. The paper systematically outlines how the GMIPS estimator is built (including special cases MSIPS, MIIPS, and MRIPS). Each estimator’s performance is measured primarily via MSE, decomposed into squared bias plus variance. This approach (a direct MSE-based comparison) is well-aligned with standard OPE practice. The datasets and synthetic setups are described thoroughly, and the results appear mostly reproducible from the provided details.

**Other Comments Or Suggestions:**

See the above weaknesses section.

**Other Strengths And Weaknesses:**

Strengths
- The authors propose an extension of embedding-based approaches (previously in single-action OPE) into the ranking domain, with a thorough theoretical backgrounds (unbiasedness, variance guarantees) under well-defined assumptions.
- The experiments demonstrate improvements on both synthetic and real data.
- They highlight the tension between embedding dimensions (for controlling bias vs. variance) and propose a pragmatic approach (SLOPE) for hyperparameter tuning.
- The writing is generally clear, with helpful notation and appendices that ensure reproducibility.

Weaknesses
- The real-world transformations rely on learning embeddings in a preprocessing step. While common, the performance can be sensitive to how the embeddings are learned (the paper does mention partial discretization, but further details or ablations on embedding-learning strategies might be interesting).
- The proposed frameworks can be mostly seen as straightforward applications of the embedding-based OPE approach and the use of SLOPE to the ranking setup, and thus the novelty is somewhat limited
- Some scenarios in real recommender systems may not fully align with the “no direct effect” assumption. The authors do discuss potential violations, but practical tips on how to mitigate or approximate are mostly in the supplemental.
- The paper relies on extreme classification data, not ranking data in their real-world experiments for no particular justification. Moreover, it would be difficult to believe that the real-world experiment is large enough as they have only 20 unique actions, which is far smaller than the real industry setup.

**Questions For Authors:**

How sensitive is GMIPS to suboptimal or incorrectly learned embeddings? Any early stopping or representation learning guidelines?

Have you ever run and evaluated the proposed approach in an industry scale problem with over thousands unique items?

Have you considered generalizing GMIPS to scalar-reward “slate” settings? Could a version of your approach with a compound embedding still reduce variance for large action spaces in slate tasks?

**Relation To Broader Scientific Literature:**

This paper fits into the broader scope of off-policy evaluation for recommender systems and slate/ranking bandits, specifically targeting the challenge of exponential growth of ranking actions. Past work on IPS-based ranking OPE includes:

- IIPS (Li et al. 2018) and RIPS (McInerney et al. 2020), which reduce variance by restricting the set of relevant actions.


- AIPS (Kiyohara et al. 2023) that adapts the user-behavior assumption to the user’s context.


- Distributional/Embedding approaches in single-action OPE (Saito & Joachims 2022), extended here to the ranking setting.

The authors properly situate their GMIPS approach as bridging these lines of work: they incorporate embedding-based marginalization (as in MIPS/embedding-based single-action OPE) while maintaining a user-behavior perspective (as in RIPS/AIPS). The references and comparisons to existing methods are appropriately detailed.

**Theoretical Claims:**

The paper provides multiple theorems:

- GMIPS unbiasedness (Proposition 3.6) under the new assumptions, even if the conventional assumption of action-level common support is not satisfied (though embedding-level support is required).


- Variance reduction (Theorem 3.7, Theorem C.1) by marginalizing on lower-dimensional embedding subsets, compared to GIPS on the full action space.


- Bias analysis (Theorem 3.8, Theorem C.3) quantifies how assumption violations (no direct effect or user-behavior-on-embeddings) contribute to systematic bias, explaining the trade-off.

The proofs in the appendix are mathematically sound on a cursory check. The notations are consistent, and the steps appear carefully reasoned. No obvious flaws are apparent in the derivations.

---

> ### Author Rebuttal · Authors · 2025-03-30
>
> We appreciate your review of our proposal paper. Below, we would like to address questions and weaknesses you have raised.
>
> >”How sensitive is GMIPS to suboptimal or incorrectly learned embeddings? Any early stopping or representation learning guidelines?”
>
> Thank you for your insightful questions. No action embeddings exist in the real-world data we utilized. Therefore, to assume that some embeddings were obtained beforehand, we learned the abstraction from the true labels in advance and used them as “embeddings”. This approach is necessary because we are not facing a scenario where the embeddings themselves are unknown; rather, we are dealing with a realistic situation in which the set of embeddings that satisfy Assumption 3.2 (No Direct Effect on Rankings) is unknown. (Please refer to the discussion with the other reviewer for more details, if you have the time.)
>
> To preface, If the available action embeddings, such as movie genres and actors, have little or no causal effect on the reward function - even in high-dimensional spaces - (which corresponds to inadequate representation learning as “a preprocessing step” in our real-world data experiments), according to Theorem 3.8, GMIPS (particularly MIIPS) suffers from significant bias. In other words, in this case, since the action embedding is equivalent to nonexistence, the MSE can be minimized by learning the optimal action representations (also referred to as embeddings) “based on the logged data”. For example, this can be achieved by integrating our Assumption 3.3 with the representation learning method proposed by (Kiyohara et al., 2024). They proposed a latent IPS (LIPS) estimator that optimizes action representations to minimize the MSE using only logged data in slate settings when the embedding itself is unknown. In contrast, If we can obtain high-quality embeddings even in low dimensions as prior information, we can apply the SLOPE algorithm, as GMIPS suffers from high variance.
>
> >”Have you ever run and evaluated the proposed approach in an industry scale problem with over thousands unique items?”
>
> Thank you for your insightful questions. The answer to this question is no; I have not. However, even if the number of unique actions increases, our estimator operates under Assumption 3.1 (Common Ranking Embedding Support), which helps reduce variance, provided that the unique action embedding does not become high-dimensional (Figure 2(b)). On the other hand, if a higher-dimensional unique embedding is required to satisfy Assumption 3.2, SLOPE can be employed to balance the trade-off between bias and variance.
>
> Additionally, I would like to address your concern that this study may not adequately tackle the large-scale problem due to the limited number of unique actions. In the ranking setting, the ranking actions, denoted as $\boldsymbol{a} \in \Pi(\mathcal{A})$, represents a unique item. We have established the number of actions per position as $|\mathcal{A}_k| = 20$ independently (to reduce the computational cost of marginalization) and the number of rankings as $K = 5$. This results in a total number of unique items in the standard OPE sense, calculated as $|\mathcal{A}| = |\mathcal{A}_k| \times K = 100$. Consequently, the total number of combinations $|\Pi(\mathcal{A})|$ is $20^5$. In contrast, the real data experiment in the existing study (Kiyohara et al., 2023) set the number of actions per position to $|\mathcal{A}_k| = 2$, and the number of rankings was $K = 6$, resulting in a total number of combinations $|\Pi(\mathcal{A})|$ equal to $2^6$. Therefore, compared to theirs, we conducted experiments on a significantly larger scale. However, you are correct that a total of 100 actions is not substantial from an industry perspective. In practice, since ranking thousands of items is challenging, for instance, a two-stage recommendation technique (Ma et al., 2020) can be employed to develop a definition of the ranking action spaces.
>
> >”Have you considered generalizing GMIPS to scalar-reward “slate” settings? Could a version of your approach with a compound embedding still reduce variance for large action spaces in slate tasks?”
>
> Thank you for your insightful questions.  Yes, it is certainly possible to apply our method to a slate bandit problem. If action embeddings can be obtained as prior information, our method can be effectively applied. For instance, by constructing an estimator that makes decisions an embedding at each slot, one would expect to reduce variance if the embedding space is smaller than the action space for each slot.
>
> ---
>
> (Ma et al., 2020) Off-Policy Learning in Two-stage Recommender Systems.
>
> (Kiyohara et al., 2023) Off-Policy Evaluation of Ranking Policies under Diverse User Behavior.
>
>  (Kiyohara et al., 2024) Off-Policy Evaluation of Slate Bandit Policies via Optimizing Abstraction.

---

### Official Review · Reviewer_HZ3E · 2025-02-27

**Overall Recommendation:** 1

**Summary:**

This paper studies off-policy evaluation (OPE) for the ranking problem. The key challenges in this setting are the length of ranking and the number of actions that may be chosen for each position. To deal with these difficulties, there are two distinct existing works. One is introducing some user behavior assumptions to reduce the effective length of ranking to consider, and the other one is using action embeddings to make OPE efficient to the number of actions. This paper proposes to combine these two approaches and show its theoretical and empirical benefits in the ranking setting.

**Claims And Evidence:**

- Overall, the paper is well-organized and claims are adequately supported by reference work, theoretical analysis, and experiments.
- Specifically, the theoretical analysis applies the Theorem 3.5 of the MIPS paper (Saito & Joachims, 2022) to the ranking setting, and gets a similar and reasonable demonstration of variance reduction.
- Experiments also show that the proposed marginalized estimator works better than its non-marginalized counterpart in a variety of user behavior assumptions, including the ones that change depending on users.

---
(Saito & Joachims, 2022) Off-Policy Evaluation for Large Action Spaces via Embeddings.

**Essential References Not Discussed:**

NA

**Experimental Designs Or Analyses:**

The experiment setting is reasonable and shows the benefit of the proposed method.
- One potential suggestion is to use a marginalized version of the AIPS estimator, but the current result already demonstrates the benefits, so this is not a strong limitation.

**Methods And Evaluation Criteria:**

- The method is reasonable, making the best of user behavior assumption and action embeddings make sense.
- However, the proposed method is somewhat incremental. The method is a naive combination of two existing approaches, and there are no eye-opening tricks to effectively combine two ways. I don't think the idea is impactful enough as an ICML paper.
- For evaluation, datasets and experiment procedures follow the standard process of OPE experiments in rankings. The use of classification datasets for real data experiments makes sense.

**Other Comments Or Suggestions:**

NA

**Other Strengths And Weaknesses:**

As already discussed, the originality is limited and the proposed method is a naive combination of two well-established ideas.

**Questions For Authors:**

- Are there further advantages of the proposed method than variance reduction enjoys (variance reduction of AIPS/GIPS) + (variance reduction of using action embeddings in AIPS/GIPS)?
- What is the advantage of the proposed method to (Kiyohara et al., 2024)?

**Relation To Broader Scientific Literature:**

- While (Kiyohara et al., 2024) consider a slightly different setting of the slate contextual bandit, the idea for OPE of slates should be applicable to OPE of rankings. They proposed to learn embeddings of entire slates when action embeddings of each slate item are available. What is the advantage of the proposed method to (Kiyohara et al., 2024)?

---
(Kiyohara et al., 2024) Off-Policy Evaluation of Slate Bandit Policies via Optimizing Abstraction.

**Theoretical Claims:**

- Comparing the theoretical analysis of the proposed estimator to a similar existing one, it seems that the conclusion is reasonable and correct (though I did not read the proofs line by line).
- However, as a similar limitation to the methodology of the paper, the finding is incremental. Theoretical analysis is simply extending the discussion of MIPS in the ranking setting, and there are no remarkable novel findings.
- It also seems that the derived variance reduction can be simply rewritten as (variance reduction of AIPS/GIPS) + (variance reduction of using action embeddings in AIPS/GIPS).

---

> ### Author Rebuttal · Authors · 2025-03-30
>
> We appreciate your review of our proposal paper. Below, we would like to address questions and weaknesses you have raised.
>
> >”Are there further advantages of the proposed method than variance reduction enjoys (variance reduction of AIPS/GIPS) + (variance reduction of using action embeddings in AIPS/GIPS)?”
>
> Thank you for your insightful comments. Our estimator offers two other advantages. First, the bias of our estimator is smaller than that of MIPS (Saito & Joachims, 2022), which is equivalent to our MIIPS, among GMIPS. Specifically, we have empirically demonstrated that our other estimator, such as MRIPS, achieved the lowest MSE  compared to MIIPS (MIPS), which exhibits bias issues in our experiments. For large action problems, it is crucial to minimize the variance of the estimator. However, as indicated by Theorem 3.8, when user behavior is complex (reflected in the first term of the bias) and the range of possible rewards is extensive (for example, when the reward is measured in video viewing time, represented by the second term of the bias), existing MIPS, which assume independent behavior on ranking embeddings, can introduce significant bias. Therefore, employing GMIPS, which accommodates complex user behavior on ranking embeddings, not only reduces variance but also mitigates bias.
>
> The second advantage is that our GMIPS will serve as the foundational technology for efficient off-policy learning (we focus on off-policy evaluation) and its extension to the OFFCEM estimator (Saito et al., 2023) in ranking settings. Our estimator enables the definition of a ranking embedding that allows for overlap between positions. This approach facilitates efficient learning without the need to consider probability distributions that do not overlap actions between positions, as is the case with the Plackett-Luce model, which incurs significant computational costs to compute the policy distribution. For instance, suppose we have a set of discrete, one-dimensional embeddings, such as movie genres, denoted as $\mathcal{E} = \lbrace e_1, e_2, e_3 \rbrace$ for each position. We can then learn a duplicate selectable policy that selects embeddings and assumes the specific user behavior. After sampling the ranking embeddings $\boldsymbol{e} = (e_2, e_2)$ where $K = 2$ from the learned policy, we can employ a regression model to determine the action associated with $e_2$ for each position (Of course, this is just one example). Furthermore, this example can be regarded as an extension of OFFCEM (Saito et al., 2023) as well as MIPS. To demonstrate the theoretical properties of OFFCEM, we must utilize the Lemma B.1 presented in (Saito & Joachims, 2022). Consequently, we consider our study to be a significant contribution that offers a robust theoretical foundation (Theorem 3.8). In other words, we believe that our Lemma C.2 is essential for developing an extension of OFFCEM for ranking purposes.
>
> As you noted, this study presents a straightforward concept based on the multiplication of (Saito & Joachims, 2022) and (Kiyohara et al., 2023). However, we believe that our research offers a robust theoretical foundation for future advancements.
>
> >”What is the advantage of the proposed method to (Kiyohara et al., 2024)?”
>
> Thank you for your insightful questions. Our study has two advantages over theirs. Our study differs slightly from theirs in terms of the problem setting. Specifically, we utilize prior information, such as movie genres and actors as “action embeddings”. In contrast, they assume a scenario in which action embeddings are unavailable and instead learn "abstractions" of actions from logged data.
>
> The first advantage is that when action embeddings are available as prior information, there is a reduced need for expensive training abstraction. This is because they requires three learners to obtain optimal (slate) abstraction. In contrast, GMIPS can achieve the best performance by selecting the best embedding through the SLOPE algorithm if we have some action embeddings.
>
> The second advantage is that our Assumption 3.3 (User Behavior Model on Ranking Embedding Spaces) is crucial for extending their abstract optimization method to ranking settings. In the slate setting, the reward is observed as a scalar value; however, in the ranking setting, the reward is represented by the number of rankings. More importantly, Since the value of the reward is influenced by the action representations taken in other positions, we can only develop an estimator that assumes independent user behavior on representation spaces without our Assumption 3.3.
>
> ---
>
> (Saito & Joachims, 2022) Off-Policy Evaluation for Large Action Spaces via Embeddings.
>
> (Saito et al., 2023) Off-Policy Evaluation for Large Action Spaces via Conjunct Effect Modeling.
>
> (Kiyohara et al., 2023) Off-Policy Evaluation of Ranking Policies under Diverse User Behavior.
>
>  (Kiyohara et al., 2024) Off-Policy Evaluation of Slate Bandit Policies via Optimizing Abstraction.

---

### Official Review · Reviewer_mFAa · 2025-03-13

**Overall Recommendation:** 2

**Summary:**

The paper studies off-policy evaluation for ranking policies. A key challenge lies in the large action spaces, which makes OPE difficult as the distributional shift between target and behavior policies become more pronounced in these settings. To address this challenge, the author(s) proposed to employ actions embeddings to alleviate the distributional shift. They further couple this approach with existing estimators that do not use embeddings, incorporating varying degrees of assumptions about user behavior.

Theoretically, the authors demonstrate that employing action embeddings reduces the variance of the resulting OPE estimator. Additionally, they provide an upper bound on the bias in cases where actions have direct effects on rewards. They further conducted empirical studies to demonstrate the advantages of their proposal over existing state-of-the-art.

**Claims And Evidence:**

The author(s) successfully demonstrated that their proposed estimator achieves smaller variance than existing estimators without action embeddings both theoretically and empirically. In theory, they provided upper bounds for the biases of their estimators without the no direct effect assumption. In simulations, they also investigated the finite-sample performance of their estimator under the violation of this assumption. The claims made in the paper were thus well-supported both theoretically and empirically.

**Essential References Not Discussed:**

I do not think there are essential references not included in the paper.

**Experimental Designs Or Analyses:**

As I have mentioned, I did not spot any evident errors in the analysis, though I did not carefully review the supplementary material.

**Methods And Evaluation Criteria:**

The environments utilized in the experiments have been adopted from previous studies, and a real-world dataset is also included. MSEs are primarily used as the evaluation criteria in numerical experiments. In theory, both variance and bias are used to quantify the performance of the estimator. To further enhance the paper, it would be beneficial to report the variance and bias of the estimators in each simulation setting.

**Other Comments Or Suggestions:**

There is a recent paper that discusses the use of state-action representation for general OPE https://proceedings.neurips.cc/paper_files/paper/2023/file/83dc5747870ea454cab25e30bef4eb8a-Paper-Conference.pdf, not tailored to the evaluation of ranking policies. It might be beneficial to discuss.

**Other Strengths And Weaknesses:**

I have discussed the strengths, novelties and contributions of the paper in earlier sections. One of my primary concern lies in the presentation. It might be beneficial to include a motivating example to better illustrate the evaluation of ranking policies and the use of embeddings. Specifically, you could start by clearly defining the contexts, actions, and rewards in the context of ranking examples. Then, when introducing the proposed methods, you could revisit this example to demonstrate how the embeddings are constructed and applied. Without such an example, the paper may come across as overly abstract for a general audience.

Another point concerns the estimation of the embeddings. If my understanding is correct, the authors assume the embeddings are known and do not elaborate on how to learn them. Firstly, this limits the methodological contribution. Secondly, while this assumption allows for the derivation of theoretical results, it potentially limits the theoretical contribution as well. Specifically, when embeddings are known and under the no-direct-effect assumption, the proposed estimator with embeddings is expected to outperform classical estimators without embeddings. Although the theoretical results derived under this assumption are valuable, they are somewhat expected given the setup.
However, in more realistic settings where prior knowledge of embeddings is unavailable, it would be highly valuable to develop methodologies for learning embeddings to reduce the cardinality of the action space. Coupled with this, developing associated theories for such settings could significantly enhance both the practical applicability and theoretical innovation of the proposed approach.

**Questions For Authors:**

Would you please give some specific examples of action embeddings if they are unknown?

**Relation To Broader Scientific Literature:**

The key contribution of the paper lies in the development of action embeddings for off-policy evaluation of ranking policies. Existing literature has not studied embeddings for evaluating ranking policies. Meanwhile, the paper also couples the idea of action embeddings with existing evaluation algorithms for ranking policies.

**Theoretical Claims:**

The theories derived in the paper seem reasonable. I did not spot any evident errors in the derivation.

---

> ### Author Rebuttal · Authors · 2025-03-30
>
> We appreciate your review of our proposal paper. Below, we would like to address questions and weaknesses you have raised.
>
> >"Would you please give some specific examples of action embeddings if they are unknown?”
>
> Thank you for your insightful question. If the action embeddings $\boldsymbol{e}$ and their distribution are unknown, we can utilize the latent representation technique (Kiyohara et al., 2024), which is the most relevant to our work, or action clustering methods (Peng et al., 2023), as demonstrated in previous studies. (Kiyohara et al., 2024) proposed a latent IPS (LIPS) estimator that optimizes action representations to minimize the MSE using only logged data in slate settings. However, their approach assumes a slate setting, where multiple actions, such as color and size of text in thumbnail recommendations, are selected, and a scalar reward, such as a click, is obtained. To effectively minimize the MSE in a ranking context, we need to develop a latent representation that accounts for the length of rankings and the variability of rewards based on user behavior across different positions. (Peng et al., 2023) propose learning the cluster function for each action using only logged data in single action decision-making.
>
> We believe that their methods can be adapted to develop our estimators in ranking setting even when the embedding itself is unknown. However, we are not analyzing scenarios where the embedding is entirely unknown; instead, we are thoroughly examining situations where some embeddings are available, but we do not know the extent to which their use will satisfy Assumption 3.2 (No Direct Effect on Rankings), which is also realistic scenario. We will explain this in more detail below.
>
> Next, we will discuss the weaknesses of the study you have provided.
>
> >”the paper may come across as overly abstract for a general audience.”
>
> I have come to understand that this is indeed true. Thank you for your insightful words; I will incorporate them into the camera-ready version.
>
> >”Another point concerns the estimation of the embeddings. If my understanding is correct, the authors assume the embeddings are known and do not elaborate on how to learn them.”
>
> Thank you for your insightful comments. I would like to clarify a point that we have thoroughly examined. As mentioned earlier, in realistic scenarios where the embedding itself is unknown (or the available embedding is of poor quality), established methods exist to estimate abstractions using only logged data. (Kiyohara et al., 2024,  Peng et al., 2023). In this study, we investigate situations where some “embeddings”, such as movie genres, as prior information are available, but we do not know the extent to which their use will satisfy Assumption 3.2. For instance, consider a 3-dimensional embedding set, $\mathcal{E} = \lbrace \mathcal{E}_1, \mathcal{E}_2, \mathcal{E}_3 \rbrace$ for each position, where we define $\mathcal{E}_1 = \text{"movie genre"}$, $\mathcal{E}_2 = \text{"actor"}$, and $\mathcal{E}_3 = \text{"location"}$ (which these can typically be obtained in practice). In this context, we now assume that Assumption 3.2 holds when we utilize only $\mathcal{E}_1$ and $\mathcal{E}_2$ for each position. This study assumes that the validity of this fact itself is unknown, even in realistic scenarios. In Section 3.3, we argued that if Assumption 3.2 does not hold, the number of embedding choices from $\mathcal{E}$ and Assumption 3.3 (User Behavior Model on Ranking Embeddings) we utilize, are the parameters that govern the bias-variance trade-off. In this case, if we employ all embedding $\mathcal{E}$ and our MSIPS, we encounter significant variance issues (Theorem 3.7 and the right side of Figure 3). Conversely, if we use only $\mathcal{E}_1$ and the MIIPS, we experience significant bias (Theorem 3.8 and the middle of Figure 3). Please note that in Figure 3, we assume that if we use a 10-dimensional embedding, Assumption 3.2 holds.
>
> No action embeddings exist in the real-world data we utilized. Therefore, to assume that some embeddings were obtained as the logged data, we learned the abstraction from the true labels in advance and used them as “embeddings” (I sincerely apologize for the absence of a detailed explanation in the paper). To address the bias-variance dilemma arising from uncertainty about which embedding satisfies Assumption 3.2, we applied the SLOPE algorithm. This algorithm automatically determines the optimal dimension of available embeddings to minimize the MSE, specifically for our MRIPS estimator. This estimator achieved the lowest MSE when sample sizes were small, as illustrated in Figure 4, where we utilized real-world data, for which we do not know whether the available embedding satisfies Assumption 3.2.
>
> ---
>
> (Kiyohara et al., 2024) Off-Policy Evaluation of Slate Bandit Policies via Optimizing Abstraction.
>
> (Peng et al., 2023) Offline Policy Evaluation in Large Action Spaces via Outcome-Oriented Action Grouping.

---

> > ### Comment · Reviewer_mFAa · 2025-04-05
> >
> > Thank you for your efforts devoted to the rebuttal. I’m still confused regarding the last point. In your theorems, the embedding set is assumed to be known. But in the real data analysis, you did learn these embeddings from data. So the theoretical framework doesn’t account for cases where embeddings are unknown or data-dependent? Is that correct?

---

> > > ### Author Response · Authors · 2025-04-06
> > >
> > > Thank you for your comments on our rebuttal. Below are our responses to your concerns.
> > >
> > > >"So the theoretical framework doesn’t account for cases where embeddings are unknown or data-dependent? Is that correct?"
> > >
> > > That is correct. Therefore, as a semi-synthetic setup, we learned embeddings from the "raw data", which contains context and label pairs.
> > >
> > > >"But in the real data analysis, you did learn these embeddings from data."
> > >
> > > We obtained embeddings of labels from raw data rather than from logged bandit data $\mathcal{D}$ through training. Consequently, we can observe logged bandit data $\mathcal{D}$ where $\boldsymbol{e}$ is observed deterministically when a ranking action $\boldsymbol{a}$ is selected by $\pi_0$. In other words, since $\boldsymbol{e}$ does not depend on the logged bandit data itself, our theoretical framework can be applied in this real-world data experiment.
> > >
> > > Please let us know if you still have any concerns.

---

### Official Review · Reviewer_41ha · 2025-03-14

**Overall Recommendation:** 3

**Summary:**

- The paper addresses the problem of off-policy evaluation (OPE) in environments with large ranking action spaces. A key challenge in this area is the high variance associated with existing estimators.

- To tackle this, the authors introduce two assumptions:
  - No Direct Effect on Rankings
  - User Behavior Model on Ranking Embedding Spaces.

- The paper proposes the Generalized Marginalized Inverse Propensity Score (GMIPS) estimator, which is designed to be unbiased while achieving variance reduction.
- The paper demonstrates that GMIPS achieves the lowest mean squared error (MSE) compared to existing methods.

**Claims And Evidence:**

Yes

**Essential References Not Discussed:**

I did not notice

**Experimental Designs Or Analyses:**

I have reviewed the benchmark experiment setup, including the details and results. Please refer to the section below for my detailed comments and observations.

**Methods And Evaluation Criteria:**

The proposed methods and evaluation criteria are reasonable.

**Other Comments Or Suggestions:**

See the section above.

**Other Strengths And Weaknesses:**

**Strengths**
- The paper introduces a novel estimator for off-policy evaluation (OPE) that addresses the high-variance challenge associated with large action spaces.
- By leveraging two key assumptions, the paper demonstrates that the new estimator is unbiased and effectively reduces variance.
- Thorough simulation studies are conducted to evaluate and understand the performance of the proposed estimator.
- The robustness of the new estimator is further validated through evaluation on a real-world case, demonstrating its practical applicability.

Comments
- Based on the experimental results, such as those presented in Figure 2, the MIIPS estimator does not show significant improvements over the snIIPS estimator compared to other variants. A more in-depth discussion on this observation would be beneficial

**Questions For Authors:**

NA

**Relation To Broader Scientific Literature:**

The paper contributes to the field of off-policy evaluation by introducing a more efficient estimator, the Generalized Marginalized Inverse Propensity Score (GMIPS), which achieves lower mean squared error (MSE) compared to existing methods. This work builds on prior research by addressing the high variance issues associated with large ranking action spaces

**Theoretical Claims:**

I did not review all the proofs for the theoretical claims presented in the paper.

---

> ### Author Rebuttal · Authors · 2025-03-30
>
> We appreciate your review of our proposal paper. Below, we would like to address weaknesses you have raised.
>
> >”Based on the experimental results, such as those presented in Figure 2, the MIIPS estimator does not show significant improvements over the snIIPS estimator compared to other variants.”
>
> Thank you for your insightful comments. MIIPS are not expected to demonstrate significant improvement due to their lower variance reduction compared to snIIPS. This can be elucidated by Theorem 3.7. Specifically, the variance reduction formula in Theorem 3.7 incorporates the variance associated with the importance weights over ranking subset spaces, given by the context and the ranking embedding subset $\mathbb{V}_{\pi_0 (\Phi_k (\boldsymbol{a}) |x, \Phi_k (\boldsymbol{e}) )}[w(x, \Phi_k (\boldsymbol{a}))]$. In other words, the greater this variance, the more significant the variance reduction our estimator can achieve compared to existing estimators. However, both snIIPS and MIIPS utilize an independent user behavior model and do not account for combinations of unique actions. This indicates that the ranking subset $\Phi_k (\boldsymbol{a}) = \boldsymbol{a}(k) \in \mathcal{A}_k $ is not extensively broad, which consequently results in minimal variance reductions. This suggests that MIIPS, a straightforward extension of MIPS (Saito & Joachims, 2022) in a single action selection problem, has limitations. Furthermore, our proposed concept of a user behavior model based on ranking embeddings (Assumption 3.3) is crucial.
>
> ---
>
> (Saito & Joachims, 2022) Off-Policy Evaluation for Large Action Spaces via Embeddings.

---

### Decision · Program_Chairs · 2025-05-01

**Decision:**

Reject

**Comment:**

This paper focuses on the off-policy evaluation of ranking policies. To handle large action spaces, the authors propose a framework of generalized marginalized inverse propensity estimator (GMIPS) based on action embeddings. Furthermore, corresponding theoretical analyses of variance and bias are provided.

The review scores are diverse (1, 2, 3, 3). The primary concern raised centered around novelty. In the discussion period, Reviewer HZ3E (who gave the score 1) provided a detailed explanation of the concerns about novelty. In brief, the underlying reasons were that the proposed method is a simple combination of existing approaches and there is a similar embedding approach in an existing paper (Kiyohara et al., 2024). Reviewers xuP4 and mFAa both agreed with the novelty concern raised by HZ3E. Additionally, Reviewer mFAa indicated that his concerns regarding the applicability of the proposed theories to unknown embeddings were not addressed. Overall, the discussion led to a negative opinion of the paper. Given the highly competitive nature of this conference, I recommend rejection. However, I encourage the authors to carefully consider the reviewers’ feedback and take necessary steps to enhance their work for future submissions.